# BASGD: Buffered Asynchronous SGD for Byzantine Learning

## Abstract

Distributed learning has become a hot research topic due to its wide application in cluster-based large-scale learning, federated learning, edge computing and so on. Most traditional distributed learning methods typically assume no failure or attack on workers. However, many unexpected cases, such as communication failure and even malicious attack, may happen in real applications. Hence, Byzantine learning (BL), which refers to distributed learning with failure or attack, has recently attracted much attention. Most existing BL methods are synchronous, which are impractical in some applications due to heterogeneous or offline workers. In these cases, asynchronous BL (ABL) is usually preferred. In this paper, we propose a novel method, called buffered asynchronous stochastic gradient descent (BASGD), for ABL. To the best of our knowledge, BASGD is the first ABL method that can resist malicious attack without storing any instances on server. Compared with those methods which need to store instances on server, BASGD takes less risk of privacy leakage. BASGD is proved to be convergent, and be able to resist failure or attack. Empirical results show that BASGD significantly outperforms vanilla ASGD and other ABL baselines when there exists failure or attack on workers.

## 1 Introduction

Due to the wide application in cluster-based large-scale learning, federated learning (Konev̌cný et al., 2016; Kairouz et al., 2019), edge computing (Shi et al., 2016) and so on, distributed learning has recently become a hot research topic (Zinkevich et al., 2010; Yang, 2013; Jaggi et al., 2014; Shamir et al., 2014; Zhang & Kwok, 2014; Ma et al., 2015; Lee et al., 2017; Lian et al., 2017; Zhao et al., 2017; Sun et al., 2018; Wangni et al., 2018; Zhao et al., 2018; Zhou et al., 2018; Yu et al., 2019a;b; Haddadpour et al., 2019). Most traditional distributed learning methods are based on stochastic gradient descent (SGD) and its variants (Bottou, 2010; Xiao, 2010; Duchi et al., 2011; Johnson & Zhang, 2013; Shalev-Shwartz & Zhang, 2013; Zhang et al., 2013; Lin et al., 2014; Schmidt et al., 2017; Zheng et al., 2017; Zhao et al., 2018), and typically assume no failure or attack on workers.

However, in real distributed learning applications with multiple networked machines (nodes), different kinds of hardware or software failure may happen. Representative failure include bit-flipping in the communication media and the memory of some workers (Xie et al., 2019). In this case, a small failure on some machines (workers) might cause a distributed learning method to fail. In addition, malicious attack should not be neglected in an open network where the manager (or server) generally has not much control on the workers, such as the cases of edge computing and federated learning. Some malicious workers may behave arbitrarily or even adversarially. Hence, *Byzantine learning* (BL), which refers to distributed learning with failure or attack, has recently attracted much attention (Diakonikolas et al., 2017; Chen et al., 2017; Blanchard et al., 2017; Alistarh et al., 2018; Damaskinos et al., 2018; Xie et al., 2019; Baruch et al., 2019; Diakonikolas & Kane, 2019).

Existing BL methods can be divided into two main categories: synchronous BL (SBL) methods and asynchronous BL (ABL) methods. In SBL methods, the learning information, such as the gradient in SGD, of all workers will be aggregated in a synchronous way. On the contrary, in ABL methods the learning information of workers will be aggregated in an asynchronous way. Existing SBL methods mainly take two different ways to achieve resilience against *Byzantine workers* which refer to those workers with failure or attack. One way is to replace the simple averaging aggregation operation with some more robust aggregation operations, such as median and trimmed-mean (Yin et al., 2018).

Krum (Blanchard et al., 2017) and ByzantinePGD (Yin et al., 2019) take this way. The other way is to filter the suspicious learning information (gradients) before averaging. Representative examples include ByzantineSGD (Alistarh et al., 2018) and Zeno (Xie et al., 2019). The advantage of SBL methods is that they are relatively simple and easy to be implemented. But SBL methods will result in slow convergence when there exist heterogeneous workers. Furthermore, in some applications like federated learning and edge computing, synchronization cannot even be performed most of the time due to the offline workers (clients or edge servers). Hence, ABL is preferred in these cases.

To the best of our knowledge, there exist only two ABL methods: Kardam (Damaskinos et al., 2018) and Zeno++ (Xie et al., 2020). Kardam introduces two filters to drop out suspicious learning information (gradients), which can still achieve good performance when the communication delay is heavy. However, when in face of malicious attack, some work finds that Kardam also drops out most correct gradients in order to filter all faulty (failure) gradients. Hence, Kardam cannot resist malicious attack (Xie et al., 2020). Zeno++ scores each received gradient, and determines whether to accept it according to the score. But Zeno++ needs to store some training instances on server for scoring. In practical applications, storing data on server will increase the risk of privacy leakage or even face legal risk. Therefore, under the general setting where server has no access to any training instances, there have not existed ABL methods to resist malicious attack.

In this paper, we propose a novel method, called buffered asynchronous stochastic gradient descent (BASGD), for ABL. The main contributions of BASGD are listed as follows:

- To the best of our knowledge, BASGD is the first ABL method that can resist malicious attack without storing any instances on server. Compared with those methods which need to store instances on server, BASGD takes less risk of privacy leakage.
- BASGD is theoretically proved to be convergent, and be able to resist failure or attack.
- Empirical results show that BASGD significantly outperforms vanilla ASGD and other ABL baselines when there exist failure or malicious attack on workers. In particular, BASGD can still converge under malicious attack, when ASGD and other ABL methods fail.

## 2 PRELIMINARY

This section presents the preliminary of this paper, including the distributed learning framework used in this paper and the definition of Byzantine worker.

### 2.1 DISTRIBUTED LEARNING FRAMEWORK

Many machine learning models, such as logistic regression and deep neural networks, can be formulated as the following finite sum optimization problem:

$$\min_{\mathbf{w} \in \mathbb{R}^d} F(\mathbf{w}) = \frac{1}{n} \sum_{i=1}^{n} f(\mathbf{w}; z_i), \tag{1}$$

where $\mathbf{w}$ is the parameter to learn, $d$ is the dimension of parameter, $n$ is the number of training instances, $f(\mathbf{w}; z_i)$ is the empirical loss on the training instance $z_i$. The goal of distributed learning is to solve the problem in (1) by designing learning algorithms based on multiple networked machines.

Although there have appeared many distributed learning frameworks, in this paper we focus on the widely used Parameter Server (PS) framework (Li et al., 2014). In a PS framework, there are several workers and one or more servers. Each worker can only communicate with server(s). There may exist more than one server in a PS framework, but for the problem of this paper servers can be logically conceived as a unity. Without loss of generality, we will assume there is only one server in this paper. Training instances are disjointedly distributed across $m$ workers. Let $\mathcal{D}_k$ denote the index set of training instances on worker_$k$, we have $\cup_{k=1}^{m} \mathcal{D}_k = \{1, 2, \ldots, n\}$ and $\mathcal{D}_k \cap \mathcal{D}_{k'} = \emptyset$ if $k \neq k'$. In this paper, we assume that server has no access to any training instances. If two instances have the same value, they are still deemed as two distinct instances. Namely, $z_i$ may equal $z_{i'}$ $(i \neq i')$.

One popular asynchronous method to solve the problem in (1) under the PS framework is ASGD (Dean et al., 2012) (see Algorithm 1 in Appendix A). In this paper, we assume each worker samples one instance for gradient computation each time, and do not separately discuss the mini-batch case.

In PS based ASGD, server is responsible for updating and maintaining the latest parameter. The number of iterations that server has already executed is used as the global logical clock of server. At the beginning, iteration number $t = 0$. Each time a SGD step is executed, $t$ will increase by $1$ immediately. The parameter after $t$ iterations is denoted as $\mathbf{w}^t$. If server sends parameters to worker_$k$ at iteration $t'$, some SGD steps may have been excuted before server receives gradient from worker_$k$ next time at iteration $t$. Thus, we define the *delay* of worker_$k$ at iteration $t$ as $\tau_k^t = t - t'$. Worker_$k$ is *heavily delayed* at iteration $t$ if $\tau_k^t > \tau_{max}$, where $\tau_{max}$ is a pre-defined non-negative constant.

## 2.2 Byzantine Worker

For workers that have sent gradients (one or more) to server at iteration $t$, we call worker_$k$ *loyal worker* if it has finished all the tasks without any fault and each sent gradient is correctly received by the server. Otherwise, worker_$k$ is called *Byzantine worker*. If worker_$k$ is a Byzantine worker, it means the received gradient from worker_$k$ is not credible, which can be an arbitrary value. In ASGD, there is one received gradient at a time. Formally, we denote the gradient received from worker_$k$ at iteration $t$ as $\mathbf{g}_k^t$. Then, we have:

$$\mathbf{g}_k^t = \begin{cases} \nabla f(\mathbf{w}^{t'}; z_i), & \text{if worker\_}k \text{ is loyal at iteration } t; \\ arbitrary\ value, & \text{if worker\_}k \text{ is Byzantine at iteration } t, \end{cases}$$

where $0 \le t' \le t$, and $i$ is randomly sampled from $\mathcal{D}_k$.

Our definition of Byzantine worker is consistent with most previous works (Blanchard et al., 2017; Xie et al., 2019; 2020). Either accidental failure or malicious attack will result in Byzantine workers.

## 3 Buffered Asynchronous SGD

In synchronous BL, gradients from all workers are received at each iteration. During this process, we can compare the gradients with each other, and then filter suspicious ones, or use more robust aggregation rules such as median and trimmed-mean for updating. However, in asynchronous BL, only one gradient is received by the server at a time. Without any training instances stored on server, it is difficult for server to identify whether a received gradient is credible or not.

In order to deal with this problem in asynchronous BL, we propose a novel method called buffered asynchronous SGD (BASGD). BASGD introduces $B$ buffers ($0 < B \le m$) on server, and the gradient used for updating parameters will be aggregated from these buffers. The detail of the learning procedure of BASGD is presented in Algorithm 2 in Appendix A. In this section, we will introduce the details of the two key components of BASGD: buffer and aggregation function.

### 3.1 Buffer

In BASGD, the $m$ workers do the same job as that in ASGD, while the updating rule on server is modified. More specifically, there are $B$ buffers ($0 < B \le m$) on server. When a gradient $\mathbf{g}$ from worker_$s$ is received, it will be temporarily stored in buffer $b$, where $b = s\ mod\ B$, as illustrated in Figure 1. Only when each buffer has stored at least one gradient, a new SGD step will be executed. Please note that no matter whether a SGD step is executed or not, the server will immediately send the latest parameters back to the worker after receiving a gradient. Hence, BASGD introduces no barrier, and is an asynchronous algorithm.

For each buffer $b$, more than one gradient may have been received at iteration $t$. We will store the average of these gradients (denoted by $\mathbf{h}_b$) in buffer $b$. Assume that there are already $(N-1)$ gradients $\mathbf{g}_1, \mathbf{g}_2, \ldots, \mathbf{g}_{N-1}$ which should be stored in buffer $b$, and $\mathbf{h}_{b(old)} = \frac{1}{N-1} \sum_{i=1}^{N-1} \mathbf{g}_i$. When the $N$-th gradient $\mathbf{g}_N$ is received, the new average value in buffer $b$ should be:

$$\mathbf{h}_{b(new)} = \tfrac{1}{N} \sum_{i=1}^{N} \mathbf{g}_i = \tfrac{N-1}{N} \cdot \mathbf{h}_{b(old)} + \tfrac{1}{N} \cdot \mathbf{g}_N.$$

This is the updating rule for each buffer $b$ when a gradient is received. We use $N_b^t$ to denote the total number of gradients stored in buffer $b$ at the $t$-th iteration. After the parameter $\mathbf{w}$ is updated, all buffers will be zeroed out at once. With the benefit of buffers, server has access to $B$ candidate gradients when updating parameter. Thus, a more reliable (robust) gradient can be aggregated from the $B$ gradients of buffers, if a proper aggregation function $Aggr(\cdot)$ is chosen.

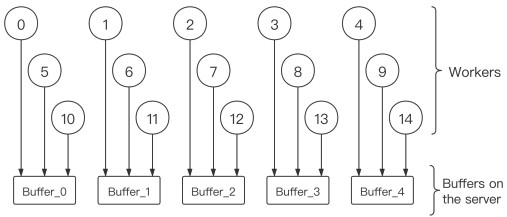

Figure 1: An example of buffers. Circle represents worker, and the number is worker ID. There are 15 workers and 5 buffers. The gradient received from worker_$s$ is stored in buffer_$\{s \bmod 5\}$.

## 3.2 AGGREGATION FUNCTION

When a SGD step is ready to be executed, there are $B$ buffers providing candidate gradients. An aggregation function is needed to get the final gradient for updating. A naive way is to take the mean of all candidate gradients. However, mean value is sensitive to outliers which are common in BL. For designing proper aggregation functions, we first define the $q$-Byzantine Robust ($q$-BR) condition to quantitatively describe the Byzantine resilience ability of an aggregation function.

**Definition 1** ($q$-Byzantine Robust). *For an aggregation function $Aggr(\cdot)$: $Aggr([\mathbf{h}_1, \ldots, \mathbf{h}_B]) = \mathbf{G}$, where $\mathbf{G} = [G_1, \ldots, G_d]^T$ and $\mathbf{h}_b = [h_{b1}, \ldots, h_{bd}]^T, \forall b \in [B]$, we call $Aggr(\cdot)$ $q$-Byzantine Robust ($q \in \mathbb{Z}, 0 < q < B/2$), if it satisfies the following two properties:*

*(a). $Aggr([\mathbf{h}_1 + \mathbf{h}', \ldots, \mathbf{h}_B + \mathbf{h}']) = Aggr([\mathbf{h}_1, \ldots, \mathbf{h}_B]) + \mathbf{h}', \ \forall \mathbf{h}_1, \ldots, \mathbf{h}_B \in \mathbb{R}^d, \forall \mathbf{h}' \in \mathbb{R}^d$;*

*(b). $\min_{s \in \mathcal{S}}\{h_{sj}\} \leq G_j \leq \max_{s \in \mathcal{S}}\{h_{sj}\}, \ \forall j \in [d], \forall \mathcal{S} \subset [B] \text{ with } |\mathcal{S}| = B - q$,*

Intuitively, property (a) in Definition 1 says that if all candidate gradients $\mathbf{h}_i$ are added by a same vector $\mathbf{h}'$, the aggregated gradient will also be added by $\mathbf{h}'$. Property (b) says that for each coordinate $j$, the aggregated value $G_j$ will be between the $(q+1)$-th smallest value and the $(q+1)$-th largest value among the $j$-th coordinates of all candidate gradients. Thus, the gradient aggregated by a $q$-BR function is insensitive to at least $q$ outliers. We can find that $q$-BR condition gets stronger when $q$ increases. In other words, if $Aggr(\cdot)$ is $q$-BR, then for any $0 < q' < q$, $Aggr(\cdot)$ is also $q'$-BR.

**Remark 1.** *It is not hard to find that when $B > 1$, mean function is not $q$-Byzantine Robust for any $q > 0$. We illustrate this by a one-dimension example: $h_1, \ldots, h_{B-1} \in [0, 1]$, and $h_B = 10 \times B$. Then $\frac{1}{B}\sum_{b=1}^{B} h_b \geq \frac{h_B}{B} = 10 \notin [0, 1]$. Namely, the mean is larger than any of the first $B - 1$ values.*

We find that the following two aggregation functions satisfy Byzantine Robust condition.

**Definition 2** (Coordinate-wise median (Yin et al., 2018)). *For candidate gradients $\mathbf{h}_1, \mathbf{h}_2, \ldots, \mathbf{h}_B \in \mathbb{R}^d$, $\mathbf{h}_b = [h_{b1}, h_{b2}, \ldots, h_{bd}]^T, \forall b = 1, 2, \ldots, B$. Coordinate-wise median is defined as:*

$$Med([\mathbf{h}_1, \ldots, \mathbf{h}_B]) = [Med(h_{\cdot 1}), \ldots, Med(h_{\cdot d})]^T,$$

*where $Med(h_{\cdot j})$ is the scalar median of the $j$-th coordinates, $\forall j = 1, 2, \ldots, d$.*

**Definition 3** (Coordinate-wise $q$-trimmed-mean (Yin et al., 2018)). *For any positive interger $q < B/2$ and candidate gradients $\mathbf{h}_1, \mathbf{h}_2, \ldots, \mathbf{h}_B \in \mathbb{R}^d$, $\mathbf{h}_b = [h_{b1}, h_{b2}, \ldots, h_{bd}]^T, \forall b = 1, 2, \ldots, B$. Coordinate-wise $q$-trimmed-mean is defined as:*

$$Trm([\mathbf{h}_1, \ldots, \mathbf{h}_B]) = [Trm(h_{\cdot 1}), \ldots, Trm(h_{\cdot d})]^T,$$

*where $Trm(h_{\cdot j})$ is the scalar $q$-trimmed-mean: $Trm(h_{\cdot j}) = \frac{1}{B-2q}\sum_{b \in \mathcal{M}_j} h_{bj}$. $\mathcal{M}_j$ is the subset of $\{h_{bj}\}_{b=1}^{B}$ obtained by removing the $q$ largest elements and $q$ smallest elements.*

In the following content, coordinate-wise median and coordinate-wise $q$-trimmed-mean are also called *median* and *trmean*, respectively. Proposition 1 shows the $q$-BR property of these two functions.

**Proposition 1.** *Coordinate-wise $q$-trmean is $q$-BR, and coordinate-wise median is $\lfloor \frac{B-1}{2} \rfloor$-BR.*

Here, $\lfloor x \rfloor$ is the maximum integer not larger than $x$. According to Proposition 1, both median and trmean are proper choices for aggregation function in BASGD. The proof can be found in Appendix B.

Now we define another class of aggregation functions, which is also important in analysis in Section 4.

**Definition 4** (Stable aggregation function). *Aggregation function $Aggr(\cdot)$ is said to be* stable *provided that $\forall \mathbf{h}_1, \ldots, \mathbf{h}_B, \tilde{\mathbf{h}}_1, \ldots, \tilde{\mathbf{h}}_B \in \mathbb{R}^d$, letting $\delta = (\sum_{b=1}^{B} \|\mathbf{h}_b - \tilde{\mathbf{h}}_b\|^2)^{\frac{1}{2}}$, we have:*

$$\|Aggr(\mathbf{h}_1, \ldots, \mathbf{h}_B) - Aggr(\tilde{\mathbf{h}}_1, \ldots, \tilde{\mathbf{h}}_B)\| \leq \delta.$$

If $Aggr(\cdot)$ is a stable aggregation function, it means that when there is a disturbance with $L_2$-norm $\delta$ on buffers, the disturbance of aggregated result will not be larger than $\delta$.

**Definition 5** (Effective aggregation function). *A stable aggregation function $Aggr(\cdot)$ is called an $(A_1, A_2)$-effective aggregation function, provided that when there are at most $r$ Byzantine workers and $\tau_k^t = 0$ for each loyal worker_k ($\forall t = 0, 1, \ldots, T - 1$), it satisfies the following two properties:*

*(i).* $\mathbb{E}[\nabla F(\mathbf{w}^t)^T \mathbf{G}_{syn}^t \mid \mathbf{w}^t] \geq \|\nabla F(\mathbf{w}^t)\|^2 - A_1, \forall \mathbf{w}^t \in \mathbb{R}^d$;
*(ii).* $\mathbb{E}[\|\mathbf{G}_{syn}^t\|^2 \mid \mathbf{w}^t] \leq (A_2)^2, \forall \mathbf{w}^t \in \mathbb{R}^d$;

*where $A_1, A_2 \in \mathbb{R}_+$ are two non-negative constants, $\mathbf{G}_{syn}^t$ is the gradient aggregated by $Aggr(\cdot)$ at the $t$-th iteration in cases without delay ($\tau_{max} = 0$).*

For different aggregation functions, constants $A_1$ and $A_2$ may differ. $A_1$ and $A_2$ are also related to loss function $F(\cdot)$, distribution of instances, buffer number $B$, maximum Byzantine worker number $r$ and so on. Inequalities (i) and (ii) in Definition 5 are two important properties in convergence proof of synchronous Byzantine learning methods. As revealed in (Yang et al., 2020), there are many existing asynchronous Byzantine learning methods. Krum, median, and trimmed-mean are proved to satisfy these two properties (Blanchard et al., 2017; Yin et al., 2018). SignSGD (Bernstein et al., 2019) can be seen as a combination of 1-bit quantization and median aggregation, while median satisfies the properties. Bulyan (Guerraoui et al., 2018) uses an existing aggregation rule to obtain a new one, and the property of Bulyan is difficult to be analyzed alone. Zeno (Xie et al., 2019) has an asynchronous version called Zeno++ (Xie et al., 2020), and it is meaningless to check the properties for Zeno.

Please note that too large $B$ will slow down the updating frequency and damage the performance, which is supported by both theoretical (in Appendix B) and empirical (in Section 5) results. In practical application, we could estimate Byzantine worker number $r$ in advance, and set $B$ to make $Aggr(\cdot)$ be $r$-BR. Specially, $B$ is suggested to be $(2r + 1)$ for median, since median is $\lfloor \frac{B-1}{2} \rfloor$-BR.

## 4 CONVERGENCE

In this section, we theoretically prove the convergence and resilience of BASGD against failure or attack. There are two main theorems. The first theorem presents a relatively loose but general bound for all $q$-BR aggregation functions. The other one presents a relatively tight bound for each distinct $(A_1, A_2)$-effective aggregation function. Since the definition of $(A_1, A_2)$-effective aggregation function is usually more difficult to verify than $q$-BR property, the general bound is also useful. Here we only present the results. Proof details are in Appendix B. We first make the following assumptions, which also have been widely used in stochastic optimization.

**Assumption 1.** *Global loss function $F(\mathbf{w})$ is bounded below: $\exists F^* \in \mathbb{R}, F(\mathbf{w}) \geq F^*, \forall \mathbf{w} \in \mathbb{R}^d$.*

**Assumption 2** (Bounded bias). *For any loyal worker, it can use locally stored training instances to estimate global gradient with bounded bias $\kappa$: $\|\mathbb{E}[\nabla f(\mathbf{w}; z_i)] - \nabla F(\mathbf{w})\| \leq \kappa, \forall \mathbf{w} \in \mathbb{R}^d$.*

**Assumption 3** (Bounded gradient). *$\nabla F(\mathbf{w})$ is bounded: $\exists D \in \mathbb{R}^+, \|\nabla F(\mathbf{w})\| \leq D, \forall \mathbf{w} \in \mathbb{R}^d$.*

**Assumption 4** (Bounded variance). *$\mathbb{E}[\|\nabla f(\mathbf{w}; z_i) - \mathbb{E}[\nabla f(\mathbf{w}; z_i) \mid \mathbf{w}]\|^2 \mid \mathbf{w}] \leq \sigma^2, \forall \mathbf{w} \in \mathbb{R}^d$.*

**Assumption 5** ($L$-smoothness). *Global loss function $F(\mathbf{w})$ is differentiable and $L$-smooth: $\|\nabla F(\mathbf{w}) - \nabla F(\mathbf{w}')\| \leq L\|\mathbf{w} - \mathbf{w}'\|, \forall \mathbf{w}, \mathbf{w}' \in \mathbb{R}^d$.*

**Remark 2.** *Please note that we do not give any assumption about convexity. The analysis in this section is suitable for both convex and non-convex models in machine learning, such as logistic regression and deep neural networks. Also, we do not give any assumption about the behavior of Byzantine workers, which may behave arbitrarily.*

Let $N^{(t)}$ be the $(q + 1)$-th smallest value in $\{N_b^t\}_{b \in [B]}$, $N_b^t$ is the total number of gradients stored in buffer $b$ at the $t$-th iteration. We define the constant $\Lambda_{B,q,r} = \frac{(B-r)\sqrt{B-r+1}}{\sqrt{(B-q-1)(q-r+1)}}$, which will appear in Lemma 1 and Lemma 2.

**Lemma 1.** *If $Aggr(\cdot)$ is q-BR, and there are at most $r$ Byzantine workers $(r \leq q)$, we have:*

$$\mathbb{E}[||\mathbf{G}^t||^2 \mid \mathbf{w}^t] \leq \Lambda_{B,q,r} d \cdot (D^2 + \sigma^2/N^{(t)}).$$

**Lemma 2.** *If $Aggr(\cdot)$ is q-BR, and the total number of heavily delayed workers and Byzantine workers is not larger than $r$ $(r \leq q)$, we have:*

$$||\mathbb{E}[\mathbf{G}^t - \nabla F(\mathbf{w}^t) \mid \mathbf{w}^t]|| \leq \Lambda_{B,q,r} d \cdot (\tau_{max} L \cdot [\Lambda_{B,q,r} d(D^2 + \sigma^2/N^{(t)})]^{\frac{1}{2}} + \sigma + \kappa).$$

**Theorem 1.** *Let $\tilde{D} = \frac{1}{T} \sum_{t=0}^{T-1}(D^2 + \sigma^2/N^{(t)})^{\frac{1}{2}}$. If $Aggr(\cdot)$ is q-BR, $B = O(r)$, and the total number of heavily delayed workers and Byzantine workers is not larger than $r$ $(r \leq q)$, set learning rate $\eta = O(\frac{1}{L\sqrt{T}})$, we have:*

$$\frac{\sum_{t=0}^{T-1} \mathbb{E}[||\nabla F(\mathbf{w}^t)||^2]}{T} \leq O\left(\frac{L[F(\mathbf{w}^0) - F^*]}{T^{\frac{1}{2}}}\right) + O\left(\frac{rd\tilde{D}}{T^{\frac{1}{2}}(q-r+1)^{\frac{1}{2}}}\right) + O\left(\frac{rDd\sigma}{(q-r+1)^{\frac{1}{2}}}\right)$$

$$+ O\left(\frac{rDd\kappa}{(q-r+1)^{\frac{1}{2}}}\right) + O\left(\frac{r^{\frac{3}{2}} LD\tilde{D}d^{\frac{3}{2}}\tau_{max}}{(q-r+1)^{\frac{3}{4}}}\right).$$

Please note that the convergence rate of vanilla ASGD is $O(T^{-\frac{1}{2}})$. Hence, Theorem 1 indicates that BASGD has a theoretical convergence rate as fast as vanilla ASGD, with an extra constant variance. The term $O(rDd\sigma(q-r+1)^{-\frac{1}{2}})$ is caused by the aggregation function, which can be deemed as a sacrifice for Byzantine resilience. The term $O(rDd\kappa(q-r+1)^{-\frac{1}{2}})$ is caused by the differences of training instances among different workers. In independent and identically distributed (i.i.d.) cases, $\kappa = 0$ and the term vanishes. The term $O(r^{\frac{3}{2}} LD\tilde{D}d^{\frac{3}{2}}\tau_{max}(q-r+1)^{-\frac{3}{4}})$ is caused by the delay, and related to parameter $\tau_{max}$. The term is also related to the buffer size. When $N_b^t$ increases, $N^{(t)}$ may increase, and thus $\tilde{D}$ will decrease. Namely, larger buffer size will result in smaller $\tilde{D}$. Besides, the factor $(q-r+1)^{-\frac{1}{2}}$ or $(q-r+1)^{-\frac{3}{4}}$ decreases as $q$ increases, and increases as $r$ increases.

Although general, the bound presented in Theorem 1 is relatively loose in high-dimensional cases, since $d$ appears in all the three extra terms. To obtain a tighter bound, we introduce Theorem 2 for BASGD with $(A_1, A_2)$-effective aggregation function (Definition 5).

**Theorem 2.** *If the total number of heavily delayed workers and Byzantine workers is not larger than $r$, $B = O(r)$, and $Aggr(\cdot)$ is an $(A_1, A_2)$-effective aggregation function in this case. Set learning rate $\eta = O(\frac{1}{\sqrt{LT}})$, and in general asynchronous cases, we have:*

$$\frac{\sum_{t=0}^{T-1} \mathbb{E}[||\nabla F(\mathbf{w}^t)||^2]}{T} \leq O\left(\frac{L^{\frac{1}{2}}[F(\mathbf{w}^0) - F^*]}{T^{\frac{1}{2}}}\right) + O\left(\frac{L^{\frac{1}{2}}\tau_{max}DA_2 r^{\frac{1}{2}}}{T^{\frac{1}{2}}}\right) + O\left(\frac{L^{\frac{1}{2}}(A_2)^2}{T^{\frac{1}{2}}}\right)$$

$$+ O\left(\frac{L^{\frac{5}{2}}(A_2)^2\tau_{max}^2 r}{T^{\frac{3}{2}}}\right) + A_1.$$

Theorem 2 indicates that if $Aggr(\cdot)$ makes a synchronous BL method converge (i.e., satisfies Definition 5), BASGD converges when using $Aggr(\cdot)$ as aggregation function. Hence, BASGD can also be seen as a technique of asynchronization. That is to say, new asynchronous methods can be obtained from synchronous ones when using BASGD. The extra constant term $A_1$ is caused by gradient bias. When there is no Byzantine workers ($r = 0$), and instances are i.i.d. across workers, letting $B = 1$ and $Aggr(\mathbf{h}_1, \ldots, \mathbf{h}_B) = Aggr(\mathbf{h}_1) = \mathbf{h}_1$, BASGD degenerates to vanilla ASGD. Under this circumstance, there is no gradient bias ($A_1 = 0$), and the extra constant term vanishes.

In general cases, Theorem 2 guarantees BASGD to find a point such that the squared $L_2$-norm of its gradient is not larger than $A_1$ (but not necessarily around a stationary point), in expectation. Please note that Assumption 3 already guarantees that gradient's squared $L_2$-norm is not larger than $D^2$. We introduce Proposition 2 to show that $A_1$ is guaranteed to be smaller than $D^2$ under a mild condition.

**Proposition 2.** *$Aggr(\cdot)$ is an $(A_1, A_2)$-effective aggregation function, and $\mathbf{G}_{syn}^t$ is aggregated by $Aggr(\cdot)$ in synchronous setting. If $\mathbb{E}[||\mathbf{G}_{syn}^t - \nabla F(\mathbf{w}^t)|| \mid \mathbf{w}^t] \leq D, \forall \mathbf{w}^t \in \mathbb{R}^d$, then $A_1 \leq D^2$.*

$\mathbf{G}_{syn}^t$ is the aggregated result of $Aggr(\cdot)$, and is a robust estimator of $\nabla F(\mathbf{w}^t)$ used for updating. Since $||\nabla F(\mathbf{w}^t)|| \leq D$, $\nabla F(\mathbf{w}^t)$ locates in a ball with radius $D$. $\mathbb{E}[||\mathbf{G}_{syn}^t - \nabla F(\mathbf{w}^t)|| \mid \mathbf{w}^t] \leq D$ means that the bias of $\mathbf{G}_{syn}^t$ is not larger than the radius $D$, which is a mild condition for $Aggr(\cdot)$.

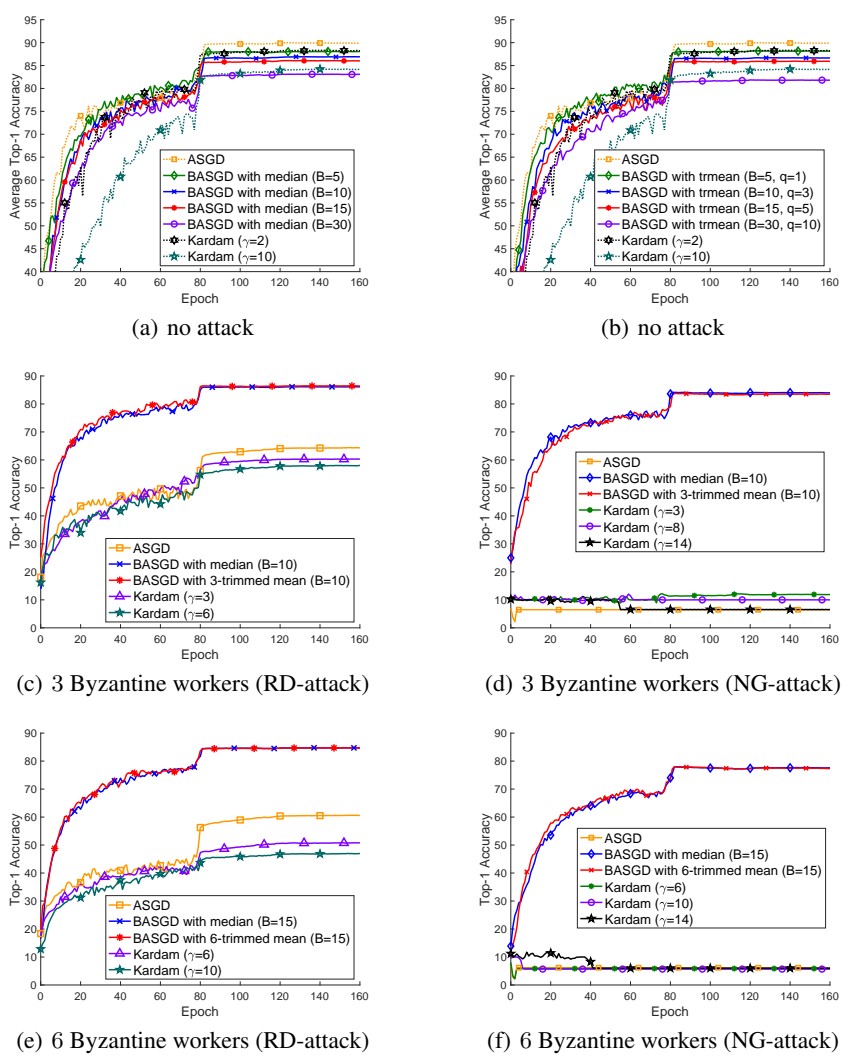

Figure 2: Average top-1 test accuracy w.r.t. epochs when there are no Byzantine workers (the first row), 3 Byzantine workers (the second row) and 6 Byzantine workers (the last row), respectively. Subfigures (c) and (e) are for RD-attack, while Subfigures (d) and (f) for NG-attack.

As many existing works have indicated (Assran et al., 2020; Nokleby et al., 2020), speed-up is also an important aspect of distributed learning methods. In BASGD, different workers can compute gradients concurrently, make each buffer be filled more quickly, and thus speed up the model updating. However, we mainly focus on Byzantine-resilience in this work. Speed-up will be thoroughly studied in future work. Besides, heavily delayed workers are considered as Byzantine in the current analysis. We will analyze heavily delayed worker's behavior more finely to obtain better results in future work.

## 5 EXPERIMENT

In this section, we empirically evaluate the performance of BASGD and baselines in both image classification (IC) and natural language processing (NLP) applications. Our experiments are conducted on a distributed platform with dockers. Each docker is bound to an NVIDIA Tesla V100 (32G) GPU (in IC) or an NVIDIA Tesla K80 GPU (in NLP). Please note that different GPU cards do not affect the reported metrics in the experiment. We choose 30 dockers as workers in IC, and 8 dockers in NLP. An extra docker is chosen as server. All algorithms are implemented with PyTorch 1.3.

Table 1: Filtered ratio of received gradients in Kardam under NG-attack (3 Byzantine workers)

| TERM | BY FREQUENCY FILTER | BY LIPSCHITZ FILTER | IN TOTAL |
|------|--------------------|--------------------|----------|
| LOYAL GRADS ($\gamma = 3$) | 10.15% (31202/307530) | 40.97% (126000/307530) | 51.12% |
| BYZANTINE GRADS ($\gamma = 3$) | 10.77% (3681/34170) | 40.31% (13773/34170) | 51.08% |
| LOYAL GRADS ($\gamma = 8$) | 28.28% (86957/307530) | 28.26% (86893/307530) | 56.53% |
| BYZANTINE GRADS ($\gamma = 8$) | 28.38% (9699/34170) | 28.06% (9588/34170) | 56.44% |
| LOYAL GRADS ($\gamma = 14$) | 85.13% (261789/307530) | 3.94% (12117/307530) | 89.07% |
| BYZANTINE GRADS ($\gamma = 14$) | 84.83% (28985/34170) | 4.26% (1455/34170) | 89.08% |

## 5.1 EXPERIMENTAL SETTING

We compare the performance of different methods under two types of attack: negative gradient attack (NG-attack) and random disturbance attack (RD-attack). Byzantine workers with NG-attack send $\tilde{\mathbf{g}}_{NG} = -k_{atk} \cdot \mathbf{g}$ to server, where $\mathbf{g}$ is the true gradient and $k_{atk} \in \mathbb{R}_+$ is a parameter. Byzantine workers with RD-attack send $\tilde{\mathbf{g}}_{RD} = \mathbf{g} + \mathbf{g}_{rnd}$ to server, where $\mathbf{g}_{rnd}$ is a random vector sampled from normal distribution $\mathcal{N}(\mathbf{0}, \|\sigma_{atk}\mathbf{g}\|^2 \cdot \mathbf{I})$. Here, $\sigma_{atk}$ is a parameter and $\mathbf{I}$ is an identity matrix. NG-attack is a typical kind of malicious attack, while RD-attack can be seen as an accidental failure with expectation $\mathbf{0}$. Besides, each worker is manually set to have a delay, which is $k_{del}$ times the computing time. Training set is randomly and equally distributed to different workers. We use the average top-1 test accuracy (in IC) or average perplexity (in NLP) on all workers w.r.t. epochs as final metrics. For BASGD, we use median and trimmed-mean as aggregation function.

Because BASGD is an ABL method, SBL methods cannot be directly compared with BASGD. The ABL method Zeno++ either cannot be directly compared with BASGD, because Zeno++ needs to store some instances on server. The number of instances stored on server will affect the performance of Zeno++ (Xie et al., 2020). Hence, we compare BASGD with ASGD and Kardam in our experiments. We set dampening function $\Lambda(\tau) = \frac{1}{1+\tau}$ for Kardam as suggested in (Damaskinos et al., 2018).

## 5.2 IMAGE CLASSIFICATION EXPERIMENT

In IC experiment, algorithms are evaluated on CIFAR-10 (Krizhevsky et al., 2009) with deep learning model ResNet-20 (He et al., 2016). Cross-entropy is used as the loss function. We set $k_{atk} = 10$ for NG-attack, and $\sigma_{atk} = 0.2$ for RD-attack. $k_{del}$ is randomly sampled from truncated standard normal distribution within $[0, +\infty)$. As suggested in (He et al., 2016), learning rate $\eta$ is set to 0.1 initially for each algorithm, and multiplied by 0.1 at the 80-th epoch and the 120-th epoch respectively. The weight decay is set to $10^{-4}$. We run each algorithm for 160 epochs. Batch size is set to 25.

Firstly, we compare the performance of different methods when there are no Byzantine workers. Experimental results with median and trmean aggregation functions are illustrated in Figure 2(a) and Figure 2(b), respectively. ASGD achieves the best performance. BASGD ($B > 1$) and Kardam have similar convergence rate to ASGD, but both sacrifice a little accuracy. Besides, the performance of BASGD gets worse when the buffer number $B$ increases, which is consistent with the theoretical results. Please note that ASGD is a degenerated case of BASGD when $B = 1$ and $Aggr(\mathbf{h}_1) = \mathbf{h}_1$. Hence, BASGD can achieve the same performance as ASGD when there is no failure or attack.

Then, for each type of attack, we conduct two experiments in which there are 3 and 6 Byzantine workers, respectively. We respectively set 10 and 15 buffers for BASGD in these two experiments. For space saving, we only present average top-1 test accuracy in Figure 2(c) and Figure 2(d) (3 Byzantine workers), and Figure 2(e) and Figure 2(f) (6 Byzantine workers). Results about training loss are in Appendix C. We can find that BASGD significantly outperforms ASGD and Kardam under both RD-attack (accidental failure) and NG-attack (malicious attack). Under the less harmful RD-attack, although ASGD and Kardam still converge, they both suffer a significant loss on accuracy. Under NG-attack, both ASGD and Kardam cannot converge, even if we have tried different values of *assumed Byzantine worker number* for Kardam, which is denoted by a hyper-parameter $\gamma$ in this paper. Hence, both ASGD and Kardam cannot resist malicious attack. On the contrary, BASGD still has a relatively good performance under both types of attack.

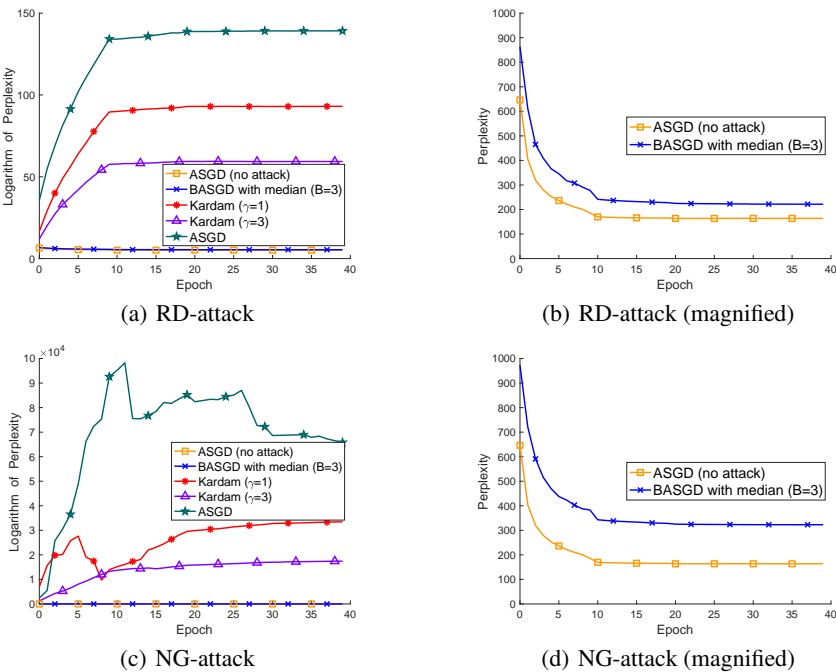

Figure 3: Average perplexity w.r.t. epochs with 1 Byzantine worker. Subfigures (a) and (b) are for RD-attack, while Subfigures (c) and (d) for NG-attack. Due to the differences in magnitude of perplexity, y-axes of Subfigures (a) and (c) are in log-scale. In addition, Subfigures (b) and (d) illustrates that BASGD converges with only a little loss in perplexity compared to the gold standard.

Moreover, we count the ratio of filtered gradients in Kardam, which is shown in Table 1. We can find that in order to filter Byzantine gradients, Kardam also filters approximately equal ratio of loyal gradients. It explains why Kardam performs poorly under malicious attack.

### 5.3 NATURAL LANGUAGE PROCESSING EXPERIMENT

In NLP experiment, the algorithms are evaluated on the WikiText-2 dataset with LSTM networks. We only use the training set and test set, while the validation set is not used in our experiment. For LSTM, we adopt 2 layers with 100 units in each. Word embedding size is set to 100, and sequence length is set to 35. Gradient clipping size is set to 0.25. Cross-entropy is used as the loss function. For each algorithm, we run each algorithm for 40 epochs. Initial learning rate $\eta$ is chosen from $\{1, 2, 5, 10, 20\}$, and is divided by 4 every 10 epochs. The best test result is adopted as the final one.

The performance of ASGD under no attack is used as gold standard. We set $k_{atk} = 10$ and $\sigma_{atk} = 0.1$. One of the eight workers is Byzantine. $k_{del}$ is randomly sampled from exponential distribution with parameter $\lambda = 1$. Each experiment is carried out for 3 times, and the average perplexity is reported in Figure 3. We can find that BASGD converges under each kind of attack, with only a little loss in perplexity compared to the gold standard (ASGD without attack). On the other hand, ASGD and Kardam both fail, even if we have set the largest $\gamma$ ($\gamma = 3$) for Kardam.

## 6 CONCLUSION

In this paper, we propose a novel method called BASGD for asynchronous Byzantine learning. To the best of our knowledge, BASGD is the first ABL method that can resist malicious attack without storing any instances on server. Compared with those methods which need to store instances on server, BASGD takes less risk of privacy leakage. BASGD is proved to be convergent, and be able to resist failure or attack. Empirical results show that BASGD significantly outperforms vanilla ASGD and other ABL baselines, when there exists failure or attack on workers.

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

---

**Algorithm 1** Asynchronous SGD (ASGD)

---

**Server:**
**Initialization:** initial parameter $\mathbf{w}^0$, learning rate $\eta$;
Send initial $\mathbf{w}^0$ to all workers;
**for** $t = 0$ **to** $t_{max} - 1$ **do**
    Wait until a new gradient $\mathbf{g}_k^t$ is received from arbitrary worker_$k$;
    Execute SGD step: $\mathbf{w}^{t+1} \leftarrow \mathbf{w}^t - \eta \cdot \mathbf{g}_k^t$;
    Send $\mathbf{w}^{t+1}$ back to worker_$k$;
**end for**
Notify all workers to stop;

**Worker_$k$:**   $(k = 0, 1, ..., m - 1)$
**repeat**
    Wait until receiving the latest parameter $\mathbf{w}$ from server;
    Randomly sample an index $i$ from $\mathcal{D}_k$;
    Compute $\nabla f(\mathbf{w}; z_i)$;
    Send $\nabla f(\mathbf{w}; z_i)$ to server;
**until** receive server's notification to stop

---

# A   ALGORITHM DETAILS

## A.1   ASYNCHRONOUS SGD (ASGD)

One popular asynchronous method to solve the problem in (1) under the PS framework is ASGD (Dean et al., 2012), which is presented in Algorithm 1.

## A.2   BUFFERED ASYNCHRONOUS SGD (BASGD)

The details of learning procedure in BASGD is presented in Algorithm 2.

# B   PROOF DETAILS

## B.1   PROOF OF PROPOSITION 1

*Proof.* Firstly, we prove coordinate-wise $q$-trimmed-mean is $q$-BR. It is not hard to check that trmean satisfies the property (a) in the definition of $q$-BR, then we prove that it also satisfies property (b).

Without loss of generality, we assume $h_{1j}, \ldots, h_{Bj}$ are already in descending order. By definition, $Trm(h_{\cdot j})$ is the average value of $\mathcal{M}_j$, which is obtained by removing $q$ largest values and $q$ smallest values of $\{h_{ij}\}_{i=1}^B$. Therefore,

$$h_{(q+1)j} = \max_{x \in \mathcal{M}_j} \{x\} \geq Trm(h_{\cdot j}) \geq \min_{x \in \mathcal{M}_j} \{x\} = h_{(n-q)j}$$

For any $\mathcal{S} \subset [B]$ with $|\mathcal{S}| = B - q$, by Pigeonhole Principle, $\mathcal{S}$ includes at least one of $h_{1j}, \ldots, h_{(q+1)j}$, and includes at least one of $h_{(n-q)j}, \ldots, h_{Bj}$. Therefore,

$$\max_{s \in \mathcal{S}} \{h_{sj}\} \geq h_{(q+1)j}; \qquad \min_{s \in \mathcal{S}} \{h_{sj}\} \leq h_{(n-q)j}.$$

Combining these two inequalities, we have:

$$\max_{s \in \mathcal{S}} \{h_{sj}\} \geq Trm(h_{\cdot j}) \geq \min_{s \in \mathcal{S}} \{h_{sj}\}.$$

Thus, coordinate-wise $q$-trimmed-mean is $q$-BR.
By definition, coordinate-wise median can be seen as $\lfloor \frac{B-1}{2} \rfloor$-trimmed-mean, and thus is $\lfloor \frac{B-1}{2} \rfloor$-BR.

$\square$

---

**Algorithm 2** Buffered Asynchronous SGD (BASGD)

---

**Server:**
**Input:** learning rate $\eta$, buffer number $B$,
    aggregation function: $Aggr(\cdot)$;
**Initialization:** initial parameter $\mathbf{w}^0$, learning rate $\eta$;
Send initial $\mathbf{w}^0$ to all workers;
Set $t \leftarrow 0$;
Set buffer: $\mathbf{h}_b \leftarrow \mathbf{0}$, $N_b^t \leftarrow 0$;
**repeat**
  Wait until receiving $\mathbf{g}$ from some worker_$s$;
  Choose buffer: $b \leftarrow s \bmod B$;
  $N_b^t \leftarrow N_b^t + 1$;
  $\mathbf{h}_b \leftarrow \frac{(N_b^t - 1)\mathbf{h}_b + \mathbf{g}}{N_b^t}$;
  **if** $N_b^t > 0$ for each $b \in [B]$ **then**
   Aggregate: $\mathbf{G}^t = Aggr([\mathbf{h}_1, \ldots, \mathbf{h}_B])$;
   Execute SGD step: $\mathbf{w}^{t+1} \leftarrow \mathbf{w}^t - \eta \cdot \mathbf{G}^t$;
   **for** $b = 1$ **to** $B$ **do**
    Zero out buffer: $\mathbf{h}_b \leftarrow \mathbf{0}$, $N_b^t \leftarrow 0$;
   **end for**
   $t \leftarrow t + 1$;
  **end if**
  Send back the latest parameters back to worker_$s$, no matter whether a SGD step is executed or not.
**until** stop criterion is satisfied
Notify all workers to stop;

 

**Worker_$k$:**   $(k = 0, 1, \ldots, m - 1)$
**repeat**
  Wait until receiving the latest parameter $\mathbf{w}$ from server;
  Randomly sample an index $i$ from $\mathcal{D}_k$;
  Compute $\nabla f(\mathbf{w}; z_i)$;
  Send $\nabla f(\mathbf{w}; z_i)$ to server;
**until** receive server's notification to stop

---

### B.2 Proof of Lemma 1

To begin with, we will introduce a lemma to estimate the ordered statistics.

**Lemma 3.** $X_1, \ldots, X_M$ are non-negative, independent and identically distributed (i.i.d.) random variables sampled from distribution $\mathcal{D}$, and have limited expectation $\mathbb{E}[X]$. Denote the $K$-th largest value in $\{X_1, \ldots, X_M\}$ as $X_{(K)}$, then $\mathbb{E}[X_{(K)}] \leq C_{M,K} \cdot \mathbb{E}[X]$, where

$$C_{M,K} = \begin{cases} M, & K = 1; \\ \frac{M!(K-1)^{K-1}(M-K)^{M-K}}{(K-1)!(M-K)!(M-1)^{M-1}}, & 1 < K < \frac{M}{2}. \end{cases}$$

*Proof.* Denote the Probability Density Function (PDF) and Cumulative Density Function (CDF) of $\mathcal{D}$ as $p(x)$ and $P(x)$, respectively. Then the PDF of $X_{(K)}$ is:

$$p_{(K)}(x) = \frac{M!}{(K-1)!(M-K)!}[1 - P(x)]^{K-1}P(x)^{M-K}p(x).$$

Thus,

$$\mathbb{E}[X_{(K)}] = \int_0^{+\infty} x \cdot p_{(K)}(x)dx$$

$$= \int_0^{+\infty} [\frac{M!}{(K-1)!(M-K)!} \cdot [1 - P(x)]^{K-1}P(x)^{M-K}] \cdot xp(x)dx$$

$$\overset{(a)}{\leq} \int_0^{+\infty} \left[ \frac{M!}{(K-1)!(M-K)!} \cdot \frac{(K-1)^{K-1}(M-K)^{M-K}}{(M-1)^{M-1}} \right] \cdot xp(x)dx$$

$$= \frac{M!(K-1)^{K-1}(M-K)^{M-K}}{(K-1)!(M-K)!(M-1)^{M-1}} \cdot \mathbb{E}[X].$$

Inequality (a) is derived based on $[1 - P(x)]^{K-1}P(x)^{M-K} \leq \frac{(K-1)^{K-1}(M-K)^{M-K}}{(M-1)^{M-1}}$, which is obtained by the following process:

Let $\theta(x) = (1-x)^{K-1}x^{M-K}$, $x \in [0,1]$.
Then $\theta'(x) = (1-x)^{K-2}x^{M-K-1}[(M-K)(1-x) - (K-1)x]$.
Let $\theta'(x) = 0$. Solving the equation, we obtain $x = \frac{M-K}{M-1}$, 0 or 1.
Also, we have $\theta(0) = \theta(1) = 0$, and $\theta(\frac{M-K}{M-1}) = \frac{(K-1)^{K-1}(M-K)^{M-K}}{(M-1)^{M-1}}$.
Then we have $\max_{x \in [0,1]} \theta(x) = \theta(\frac{M-K}{M-1}) = \frac{(K-1)^{K-1}(M-K)^{M-K}}{(M-1)^{M-1}}$.
Thus, $[1 - P(x)]^{K-1}P(x)^{M-K} = \theta(P(x)) \leq \frac{(K-1)^{K-1}(M-K)^{M-K}}{(M-1)^{M-1}}$.

$\square$

**Proposition 3.** $\forall B, q, r \in \mathbb{Z}_+, 0 \leq r \leq q < \frac{B}{2}$,

$$C_{B-r,q-r+1} \leq \frac{(B-r)\sqrt{B-r+1}}{\sqrt{(B-q-1)(q-r+1)}}.$$

*Proof.* By Stirling's approximation, we have:

$$\sqrt{2\pi n} \cdot n^n e^{-n} \leq n! \leq e\sqrt{n} \cdot n^n e^{-n}, \; \forall n \in \mathbb{Z}_+.$$

Therefore,

$$\sqrt{2\pi n} \cdot e^{-n} \leq \frac{n!}{n^n} \leq e\sqrt{n} \cdot e^{-n}, \; \forall n \in \mathbb{Z}_+. \tag{2}$$

By definition of $C_{M,k}$,

$$\begin{aligned}
C_{M,K} &= \frac{M!(K-1)^{K-1}(M-K)^{M-K}}{(K-1)!(M-K)!(M-1)^{M-1}} \\
&= M \cdot \frac{(M-1)!}{(M-1)^{M-1}} \cdot \frac{(K-1)^{K-1}}{(K-1)!} \cdot \frac{(M-K)^{M-K}}{(M-K)!} \\
&\leq M \cdot [e\sqrt{M-1} \cdot e^{-(M-1)}] \cdot \frac{e^{K-1}}{\sqrt{2\pi(K-1)}} \cdot \frac{e^{M-K}}{\sqrt{2\pi(M-K)}} \\
&= \frac{e}{2\pi} \cdot \frac{M\sqrt{M-1}}{\sqrt{(M-K)(K-1)}},
\end{aligned}$$

where the inequality uses Inequality (2).

Case (i). When $r < q$,

$$\begin{aligned}
C_{B-r,q-r+1} &\leq \frac{e}{2\pi} \cdot \frac{(B-r)\sqrt{B-r-1}}{\sqrt{(B-q-1)(q-r)}} \\
&\leq \frac{(B-r)\sqrt{B-r+1}}{\sqrt{(B-q-1)(q-r+1)}}.
\end{aligned}$$

Case (ii). When $r = q$, by definition of $C_{M,K}$, we have:

$$C_{B-r,q-r+1} = C_{B-q,1} = B - q = \frac{(B-r)\sqrt{B-r+1}}{\sqrt{(B-q-1)(q-r+1)}}.$$

In conclusion, when $r \leq q$, we have:

$$C_{B-r,q-r+1} \leq \frac{(B-r)\sqrt{B-r+1}}{\sqrt{(B-q-1)(q-r+1)}}.$$

$\square$

When $B$ and $q$ are fixed, the upper bound of $C_{B-r,q-r+1}$ will increase when $r$ (number of Byzantine workers) increases. Namely, the upper bound will be larger if there are more Byzantine workers. When $B$ and $r$ are fixed, $q$ measures the Byzantine Robust degree of aggregation function $Aggr(\cdot)$. The factor $[(B-q-1)(q-r)]^{-\frac{1}{2}}$ is monotonically decreasing with respect to $q$, when $q < \frac{B-1+r}{2}$. Since $r \le q < \frac{B}{2}$, the upper bound will decrease when $q$ increases. Also, $B-q$ decreases when $q$ increases. Namely, the upper bound will be smaller if $Aggr(\cdot)$ has a stronger $q$-BR property.

In the worst case ($q = r$), the upper bound of $C_{B-r,q-r+1}$ is linear to $B$. Even in the best case ($r = 0, q = \lfloor \frac{B-1}{2} \rfloor$), the denominator is about $\frac{B}{2}$ and the upper bound of $C_{B-r,q-r+1}$ is linear to $\sqrt{B}$. Thus, larger $B$ might result in larger error. Hence, buffer number is not supposed to be set too large.

Now we prove Lemma 1.

*Proof.*

$$\mathbb{E}[||\mathbf{G}^t||^2 \mid \mathbf{w}^t]$$
$$= \mathbb{E}[||Aggr([\mathbf{h}_1, \ldots, \mathbf{h}_B])||^2 \mid \mathbf{w}^t]$$
$$= \sum_{j=1}^{d} \mathbb{E}[Aggr([\mathbf{h}_1, \ldots, \mathbf{h}_B])_j^2 \mid \mathbf{w}^t],$$

where $Aggr([\mathbf{h}_1, \ldots, \mathbf{h}_B])_j$ represents the $j$-th coordinate of the aggregated gradient.

We use $\mathcal{H}^t$ to denote the credible buffer index set, which is composed by the index of buffers, where the stored gradients are all from loyal workers.

For each $b \in \mathcal{H}^t$, $\mathbf{h}_b$ has stored $N_b^t$ gradients at iteration $t$: $\mathbf{g}_1, \ldots, \mathbf{g}_{N_b^t}$, and we have:

$$\mathbf{h}_b = \frac{1}{N_b^t} \sum_{i=1}^{N_b^t} \mathbf{g}_i.$$

Then,

$$\mathbb{E}[||\mathbf{h}_b||^2 \mid \mathbf{w}^t] = \mathbb{E}[||\mathbf{h}_b - \mathbb{E}[\mathbf{h}_b \mid \mathbf{w}^t]||^2 \mid \mathbf{w}^t] + ||\mathbb{E}[\mathbf{h}_b \mid \mathbf{w}^t]||^2$$

$$= \mathbb{E}[||\frac{1}{N_b^t} \sum_{i=1}^{N_b^t} (\mathbf{g}_i - \mathbb{E}[\mathbf{g}_i \mid \mathbf{w}^t])||^2 \mid \mathbf{w}^t] + ||\mathbb{E}[\frac{1}{N_b^t} \sum_{i=1}^{N_b^t} \mathbf{g}_i \mid \mathbf{w}^t]||^2$$

$$\overset{(a)}{\le} \frac{\sigma^2}{N_b^t} + ||\mathbb{E}[\frac{1}{N_b^t} \sum_{i=1}^{N_b^t} \mathbf{g}_i \mid \mathbf{w}^t]||^2$$

$$= \frac{\sigma^2}{N_b^t} + \frac{1}{(N_b^t)^2} ||\sum_{i=1}^{N_b^t} \mathbb{E}[\mathbf{g}_i \mid \mathbf{w}^t]||^2$$

$$\overset{(b)}{\le} \frac{\sigma^2}{N_b^t} + \frac{1}{(N_b^t)^2} \cdot N_b^t \cdot \sum_{i=1}^{N_b^t} ||\mathbb{E}[\mathbf{g}_i \mid \mathbf{w}^t]||^2$$

$$\overset{(c)}{\le} \frac{\sigma^2}{N_b^t} + D^2.$$

Inequality (a) is derived based on Assumption 4 and the fact that $\mathbf{g}_i$ is mutually uncorrelated. Inequality (b) is derived by the following process:

$$||\sum_{i=1}^{N_b^t} \mathbb{E}[\mathbf{g}_i \mid \mathbf{w}^t]||^2 = \sum_{i=1}^{N_b^t} ||\mathbb{E}[\mathbf{g}_i \mid \mathbf{w}^t]||^2 + \sum_{1 \le i < i' \le N_b^t} 2 \cdot \mathbb{E}[\mathbf{g}_i \mid \mathbf{w}^t]^T \mathbb{E}[\mathbf{g}'_i \mid \mathbf{w}^t]$$

$$\le \sum_{i=1}^{N_b^t} ||\mathbb{E}[\mathbf{g}_i \mid \mathbf{w}^t]||^2 + \sum_{1 \le i < i' \le N_b^t} (||\mathbb{E}[\mathbf{g}_i \mid \mathbf{w}^t]||^2 + ||\mathbb{E}[\mathbf{g}'_i \mid \mathbf{w}^t]||^2$$

$$= \sum_{i=1}^{N_b^t} \|\mathbb{E}[\mathbf{g}_i \mid \mathbf{w}^t]\|^2 + (N_b^t - 1) \cdot \sum_{i=1}^{N_b^t} \|\mathbb{E}[\mathbf{g}_i \mid \mathbf{w}^t]\|^2$$

$$= N_b^t \cdot \sum_{i=1}^{N_b^t} \|\mathbb{E}[\mathbf{g}_i \mid \mathbf{w}^t]\|^2.$$

Inequality (c) is derived based on Assumption 3.

Because there are no more than $r$ Byzantine workers at iteration $t$, no more than $r$ buffers contain Byzantine gradient. Thus, the credible buffer index set $\mathcal{H}^t$ has at least $(B - r)$ elements. In case that $\mathcal{H}^t$ has more than $(B - r)$ elements, we take the indices of the smallest $(B - q)$ elements in $\{h_{bj}\}_{b \in \mathcal{H}^t}$ to compose $\mathcal{H}_j^t$, and we have $|\mathcal{H}_j^t| = B - q$.

Note that $Aggr(\cdot)$ is $q$-BR, and by definition we have:

$$\min_{b \in \mathcal{H}_j^t} \{h_{bj}\} \leq Aggr([\mathbf{h}_1, \ldots, \mathbf{h}_B])_j \leq \max_{b \in \mathcal{H}_j^t} \{h_{bj}\}.$$

Therefore,

$$\sum_{j=1}^{d} \mathbb{E}[Aggr([\mathbf{h}_1, \ldots, \mathbf{h}_B])_j^2 | \mathbf{w}^t] \leq \sum_{j=1}^{d} \mathbb{E}[\max_{b \in \mathcal{H}_j^t} \{h_{bj}^2\} | \mathbf{w}^t].$$

There are $(B - r)$ credible buffers, and we choose the smallest $(B - q)$ buffers to compose $\mathcal{H}_j^t$. Therefore, for all $b \in \mathcal{H}_j^t$, $h_{bj}$ is not larger than the $(q - r + 1)$-th largest one in $\{h_{bj}\}_{b \in \mathcal{H}^t}$. Let $N^{(t)}$ be the $(q + 1)$-th smallest value in $\{N_b^t\}_{b \in [B]}$. Using Lemma 3, we have:

$$\mathbb{E}[\max_{b \in \mathcal{H}_j^t} \{h_{bj}^2\} | \mathbf{w}^t] \leq \mathbb{E}[\max_{b \in \mathcal{H}_j^t} \{\|\mathbf{h}_b\|^2\} | \mathbf{w}^t]$$

$$\leq \mathbb{E}[\max_{b \in \mathcal{H}_j^t} \{D^2 + \frac{\sigma^2}{N_b^t}\} | \mathbf{w}^t]$$

$$= C_{B-r,q-r+1} \cdot (D^2 + \frac{\sigma^2}{N^{(t)}}).$$

Thus,

$$\mathbb{E}[\|\mathbf{G}^t\|^2 \mid \mathbf{w}^t] \leq \sum_{j=1}^{d} \mathbb{E}[\max_{b \in \mathcal{H}_j^t} \{h_{bj}^2\} | \mathbf{w}^t] \leq C_{B-r,q-r+1} d \cdot (D^2 + \frac{\sigma^2}{N^{(t)}}).$$

By Proposition 3, we have:

$$\mathbb{E}[\|\mathbf{G}^t\|^2 \mid \mathbf{w}^t] \leq d \cdot \frac{(B - r)\sqrt{B - r + 1}}{\sqrt{(B - q - 1)(q - r + 1)}} \cdot (D^2 + \frac{\sigma^2}{N^{(t)}}).$$

$\square$

### B.3 PROOF OF LEMMA 2

*Proof.*

$$\mathbb{E}[\mathbf{G}^t - \nabla F(\mathbf{w}^t) \mid \mathbf{w}^t]$$
$$= \mathbb{E}[Aggr([\mathbf{h}_1, \ldots, \mathbf{h}_B]) - \nabla F(\mathbf{w}^t) \mid \mathbf{w}^t]$$
$$= \mathbb{E}[Aggr([\mathbf{h}_1 - \nabla F(\mathbf{w}^t), \ldots, \mathbf{h}_B - \nabla F(\mathbf{w}^t)]) \mid \mathbf{w}^t], \tag{3}$$

where the second equation is derived based on the Property (b) in the definition of $q$-BR.

For each $b \in \mathcal{H}^t$, $\mathbf{h}_b$ has stored $N_b^t$ gradients at iteration $t$: $\mathbf{g}_1, \ldots, \mathbf{g}_{N_b^t}$, and we have:

$$\mathbf{h}_b - \nabla F(\mathbf{w}^t) = \frac{1}{N_b^t} \sum_{k=1}^{N_b^t} \mathbf{g}_i - \nabla F(\mathbf{w}^t) = \frac{1}{N_b^t} \sum_{k=1}^{N_b^t} [\nabla f(\mathbf{w}^{t_k}; z_{i_k}) - \nabla F(\mathbf{w}^t)],$$

where $0 \leq t - t_k \leq \tau_{max}, \forall k = 1, 2, \ldots, N_b^t$.

Taking expectation on both sides, we have:

$$\mathbb{E}[||\mathbf{h}_b - \nabla F(\mathbf{w}^t)|| \, |\mathbf{w}^t]$$

$$= \mathbb{E}[|| \frac{1}{N_b^t} \sum_{k=1}^{N_b^t} (\nabla f(\mathbf{w}^{t_k}; z_{i_k}) - \nabla F(\mathbf{w}^t))|| \, |\mathbf{w}^t]$$

$$\leq \frac{1}{N_b^t} \sum_{k=1}^{N_b^t} \mathbb{E}[||\nabla f(\mathbf{w}^{t_k}; z_{i_k}) - \nabla F(\mathbf{w}^t)|| \, |\mathbf{w}^t]$$

$$\overset{(a)}{\leq} \frac{1}{N_b^t} \sum_{k=1}^{N_b^t} \{ \mathbb{E}[||\nabla F(\mathbf{w}^{t_k}) - \nabla F(\mathbf{w}^t)|| \, |\mathbf{w}^t]$$

$$+ \mathbb{E}[||\nabla f(\mathbf{w}^{t_k}; z_{i_k}) - \mathbb{E}[\nabla f(\mathbf{w}^{t_k}; z_{i_k})]|| \, |\mathbf{w}^t]$$

$$+ \mathbb{E}[||\mathbb{E}[\nabla f(\mathbf{w}^{t_k}; z_{i_k})] - \nabla F(\mathbf{w}^{t_k})|| \, |\mathbf{w}^t] \},$$

where (a) is derived based on Triangle Inequality.

The first part:

$$\mathbb{E}[||\nabla F(\mathbf{w}^{t_k}) - \nabla F(\mathbf{w}^t)|| \, |\mathbf{w}^t]$$

$$\overset{(b)}{\leq} L \cdot \mathbb{E}[||\mathbf{w}^{t_k} - \mathbf{w}^t|| \, |\mathbf{w}^t]$$

$$= L \cdot \mathbb{E}[|| \sum_{t'=t_k}^{t-1} \mathbf{G}^{t'}|| \, |\mathbf{w}^t]$$

$$\leq \sum_{t'=t_k}^{t-1} L \cdot \mathbb{E}[||\mathbf{G}^{t'}|| \, |\mathbf{w}^t]$$

$$= \sum_{t'=t_k}^{t-1} L \cdot \sqrt{\mathbb{E}[||\mathbf{G}^{t'}|| \, |\mathbf{w}^t]^2}$$

$$\leq \sum_{t'=t_k}^{t-1} L \cdot \sqrt{\mathbb{E}[||\mathbf{G}^{t'}||^2 \, |\mathbf{w}^t]}$$

$$\overset{(c)}{\leq} \sum_{t'=t_k}^{t-1} L \cdot \sqrt{C_{B-r,q-r+1} d \cdot (D^2 + \sigma^2/N^{(t)})}$$

$$\overset{(d)}{\leq} \tau_{max} L \cdot \sqrt{C_{B-r,q-r+1} d \cdot (D^2 + \sigma^2/N^{(t)})},$$

where (b) is derived based on Assumption 5, (c) is derived based on Lemma 1 and (d) is derived based on $t - t_k \leq \tau_{max}$.

The second part:

$$\mathbb{E}[||\nabla f(\mathbf{w}^{t_k}; z_{i_k}) - \mathbb{E}[\nabla f(\mathbf{w}^{t_k}; z_{i_k})]|| \, |\mathbf{w}^t]$$

$$= \sqrt{\mathbb{E}[||\nabla f(\mathbf{w}^{t_k}; z_{i_k}) - \mathbb{E}[\nabla f(\mathbf{w}^{t_k}; z_{i_k})]|| \, |\mathbf{w}^t]^2}$$

$$\leq \sqrt{\mathbb{E}[||\nabla f(\mathbf{w}^{t_k}; z_{i_k}) - \mathbb{E}[\nabla f(\mathbf{w}^{t_k}; z_{i_k})]||^2 \, |\mathbf{w}^t]}$$

$$\overset{(e)}{\leq} \sigma,$$

where (e) is derived based on Assumption 4.

By Assumption 2, we have the following estimation for the third part:

$$\mathbb{E}[||\mathbb{E}[\nabla f(\mathbf{w}^{t_k}; z_{i_k})] - \nabla F(\mathbf{w}^{t_k})|| \, |\mathbf{w}^t] \leq \kappa.$$

Therefore,

$$\mathbb{E}[||\mathbf{h}_b - \nabla F(\mathbf{w}^t)|| \, |\mathbf{w}^t]$$

$$\leq \frac{1}{N_b^t} \sum_{k=1}^{N_b^t} (\tau_{max} L \sqrt{C_{B-r,q-r+1} d \cdot (D^2 + \sigma^2/N^{(t)})} + \sigma + \kappa)$$

$$= \tau_{max} L \sqrt{C_{B-r,q-r+1} d \cdot (D^2 + \sigma^2/N^{(t)})} + \sigma + \kappa. \tag{4}$$

Similar to the proof of Lemma 1, $\forall j \in [d]$, we have:

$$\min_{b \in \mathcal{H}_j^t} \{h_{bj} - \nabla F(\mathbf{w}^t)_j\}$$

$$\leq Aggr([\mathbf{h}_1 - \nabla F(\mathbf{w}^t), \dots, \mathbf{h}_B - \nabla F(\mathbf{w}^t)])_j$$

$$\leq \max_{b \in \mathcal{H}_j^t} \{h_{bj} - \nabla F(\mathbf{w}^t)_j\},$$

where $\mathcal{H}_j^t$ is composed by the indices of the smallest $(B-q)$ elements in $\{h_{bj} - \nabla F(\mathbf{w}^t)_j\}_{b \in \mathcal{H}^t}$. Therefore,

$$||\mathbb{E}[Aggr([\mathbf{h}_1 - \nabla F(\mathbf{w}^t), \dots, \mathbf{h}_B - \nabla F(\mathbf{w}^t)]) \mid \mathbf{w}^t]||$$

$$\leq \sum_{j=1}^d ||\mathbb{E}[Aggr([\mathbf{h}_1 - \nabla F(\mathbf{w}^t), \dots, \mathbf{h}_B - \nabla F(\mathbf{w}^t)])_j \mid \mathbf{w}^t]||$$

$$\leq \sum_{j=1}^d \mathbb{E}[||Aggr([\mathbf{h}_1 - \nabla F(\mathbf{w}^t), \dots, \mathbf{h}_B - \nabla F(\mathbf{w}^t)])_j|| \mid \mathbf{w}^t]$$

$$\overset{(f)}{\leq} \sum_{j=1}^d \mathbb{E}[\max_{b \in \mathcal{H}_j^t} ||h_{bj} - \nabla F(\mathbf{w}^t)_j|| \mid \mathbf{w}^t]$$

$$\overset{(g)}{\leq} \sum_{j=1}^d C_{B-r,q-r+1} \mathbb{E}[||h_{bj} - \nabla F(\mathbf{w}^t)_j|| \, |\mathbf{w}^t]$$

$$\leq \sum_{j=1}^d C_{B-r,q-r+1} \mathbb{E}[||\mathbf{h}_b - \nabla F(\mathbf{w}^t)|| \, |\mathbf{w}^t]$$

$$\overset{(h)}{\leq} \sum_{j=1}^d C_{B-r,q-r+1} \cdot (\tau_{max} L \sqrt{C_{B-r,q-r+1} d \cdot (D^2 + \sigma^2/N^{(t)})} + \sigma + \kappa)$$

$$= C_{B-r,q-r+1} d \cdot (\tau_{max} L \sqrt{C_{B-r,q-r+1} d \cdot (D^2 + \sigma^2/N^{(t)})} + \sigma + \kappa), \tag{5}$$

where (f) is derived based on definition of $q$-BR, (g) is derived based on Lemma 3, and (h) is derived based on Inequality (4).

Combining Equation (3) and Inequality (5), we obtain:

$$||\mathbb{E}[\mathbf{G}^t - \nabla F(\mathbf{w}^t) \mid \mathbf{w}^t]|| \leq C_{B-r,q-r+1} d \cdot (\tau_{max} L \sqrt{C_{B-r,q-r+1} d \cdot (D^2 + \sigma^2/N^{(t)})} + \sigma + \kappa).$$

By Proposition (3), we have:

$$||\mathbb{E}[\mathbf{G}^t - \nabla F(\mathbf{w}^t) \mid \mathbf{w}^t]|| \leq \frac{d(B-r)\sqrt{B-r+1}}{\sqrt{(B-q-1)(q-r+1)}}$$

$$\cdot (\tau_{max} L \sqrt{d \frac{(B-r)\sqrt{B-r+1}}{\sqrt{(B-q-1)(q-r+1)}} \cdot (D^2 + \sigma^2/N^{(t)})} + \sigma + \kappa).$$

$$\square$$

### B.4 PROOF OF THEOREM 1

*Proof.*

$$
\begin{aligned}
\mathbb{E}[F(\mathbf{w}^{t+1}) \mid \mathbf{w}^t] =& \mathbb{E}[F(\mathbf{w}^t - \eta \cdot \mathbf{G}^t) \mid \mathbf{w}^t] \\
\overset{(a)}{\leq}& \mathbb{E}[F(\mathbf{w}^t) - \eta \cdot \nabla F(\mathbf{w}^t)^T \mathbf{G}^t + \frac{L}{2}\eta^2||\mathbf{G}^t||^2 \mid \mathbf{w}^t] \\
=& F(\mathbf{w}^t) - \eta \cdot \mathbb{E}[\nabla F(\mathbf{w}^t)^T \mathbf{G}^t \mid \mathbf{w}^t] + \frac{\eta^2 L}{2}\mathbb{E}[||\mathbf{G}^t||^2 \mid \mathbf{w}^t] \\
=& F(\mathbf{w}^t) - \eta \cdot \nabla F(\mathbf{w}^t)^T \mathbb{E}[\mathbf{G}^t \mid \mathbf{w}^t] + \frac{\eta^2 L}{2}\mathbb{E}[||\mathbf{G}^t||^2 \mid \mathbf{w}^t] \\
=& F(\mathbf{w}^t) - \eta \cdot \nabla F(\mathbf{w}^t)^T \nabla F(\mathbf{w}^t) + \frac{\eta^2 L}{2}\mathbb{E}[||\mathbf{G}^t||^2 \mid \mathbf{w}^t] \\
& - \eta \cdot \nabla F(\mathbf{w}^t)^T \mathbb{E}[\mathbf{G}^t - \nabla F(\mathbf{w}^t) \mid \mathbf{w}^t] \\
\leq& F(\mathbf{w}^t) - \eta \cdot ||\nabla F(\mathbf{w}^t)||^2 + \frac{\eta^2 L}{2}\mathbb{E}[||\mathbf{G}^t||^2 \mid \mathbf{w}^t] \\
& + \eta \cdot ||\nabla F(\mathbf{w}^t)|| \cdot ||\mathbb{E}[\mathbf{G}^t - \nabla F(\mathbf{w}^t) \mid \mathbf{w}^t]||,
\end{aligned}
$$

where (a) is derived based on Assumption 5.

Using Lemma 1 and Lemma 2, we have:

$$
\begin{aligned}
& \mathbb{E}[F(\mathbf{w}^{t+1}) \mid \mathbf{w}^t] \\
\leq& F(\mathbf{w}^t) - \eta \cdot ||\nabla F(\mathbf{w}^t)||^2 + \frac{\eta^2 L}{2}C_{B-r,q-r+1}d \cdot (D^2 + \sigma^2/N^{(t)}) \\
& + \eta \cdot C_{B-r,q-r+1}d \cdot (\tau_{max}L\sqrt{C_{B-r,q-r+1}d \cdot (D^2 + \sigma^2/N^{(t)})} + \sigma + \kappa) \cdot ||\nabla F(\mathbf{w}^t)||.
\end{aligned}
$$

Also, by Assumption 3, $||\nabla F(\mathbf{w}^t)|| \leq D$.

Taking total expectation and combining $||\nabla F(\mathbf{w}^t)|| \leq D$, we have:

$$
\begin{aligned}
\mathbb{E}[F(\mathbf{w}^{t+1})] \leq& \mathbb{E}[F(\mathbf{w}^t)] - \eta \cdot \mathbb{E}[||\nabla F(\mathbf{w}^t)||^2] + \frac{\eta^2 L}{2}C_{B-r,q-r+1}d \cdot (D^2 + \sigma^2/N^{(t)}) \\
& + \eta \cdot C_{B-r,q-r+1}Dd(\tau_{max}L\sqrt{C_{B-r,q-r+1}d \cdot (D^2 + \sigma^2/N^{(t)})} + \sigma + \kappa).
\end{aligned}
$$

Let $\tilde{D} = \frac{1}{T}\sum_{t=0}^{T-1}\sqrt{D^2 + \sigma^2/N^{(t)}}$. By telescoping, we have:

$$
\begin{aligned}
\eta \cdot \sum_{t=0}^{T-1}\mathbb{E}[||\nabla F(\mathbf{w}^t)||^2] \leq& \{F(\mathbf{w}^0) - \mathbb{E}[F(\mathbf{w}^T)]\} \\
& + \eta^2 T \cdot \frac{L}{2}C_{B-r,q-r+1}d \cdot \frac{1}{T}\sum_{t=0}^{T-1}(D^2 + \sigma^2/N^{(t)}) \\
& + \eta T \cdot C_{B-r,q-r+1}Dd(\tau_{max}L\tilde{D}\sqrt{C_{B-r,q-r+1}d} + \sigma + \kappa).
\end{aligned}
$$

Note that $\mathbb{E}[F(\mathbf{w}^T)] \geq F^*$, and let $\eta = O\left(\frac{1}{L\sqrt{T}}\right)$:

$$
\begin{aligned}
\frac{\sum_{t=0}^{T-1}\mathbb{E}[||\nabla F(\mathbf{w}^t)||^2]}{T} \leq& O\left(\frac{L[F(\mathbf{w}^0) - F^*]}{\sqrt{T}}\right) + O\left(\frac{C_{B-r,q-r+1}\tilde{D}d}{\sqrt{T}}\right) \\
& + O\left(C_{B-r,q-r+1}Dd \cdot (\tau_{max}L\tilde{D}\sqrt{C_{B-r,q-r+1}d} + \sigma + \kappa)\right).
\end{aligned}
$$

When $q = r$ and $B = O(r)$, we have $C_{B-r,q-r+1} \leq \frac{(B-r)\sqrt{B-r+1}}{\sqrt{(B-q-1)(q-r+1)}} = O\left(\frac{r}{(q-r+1)^{\frac{1}{2}}}\right)$. Thus,

$$
\frac{\sum_{t=0}^{T-1}\mathbb{E}[||\nabla F(\mathbf{w}^t)||^2]}{T} \leq O\left(\frac{L[F(\mathbf{w}^0) - F^*]}{T^{\frac{1}{2}}}\right) + O\left(\frac{rd\tilde{D}}{T^{\frac{1}{2}}(q-r+1)^{\frac{1}{2}}}\right) + O\left(\frac{rDd\sigma}{(q-r+1)^{\frac{1}{2}}}\right)
$$

$$+ O\left(\frac{rDd\kappa}{(q-r+1)^{\frac{1}{2}}}\right) + O\left(\frac{r^{\frac{3}{2}} LD\tilde{D}d^{\frac{3}{2}}\tau_{max}}{(q-r+1)^{\frac{3}{4}}}\right).$$

$\square$

### B.5  Proof of Theorem 2

*Proof.* Let $\mathbf{h}_b'$ be the value of the $b$-th buffer, if all received loyal gradients were computed based on $\mathbf{w}^t$. Note $\mathbf{G}^t = Aggr(\mathbf{h}_1, \ldots, \mathbf{h}_B)$.

$$\mathbb{E}[F(\mathbf{w}^{t+1}) \mid \mathbf{w}^t]$$
$$= \mathbb{E}[F(\mathbf{w}^t - \eta \cdot \mathbf{G}^t) \mid \mathbf{w}^t]$$
$$\overset{(a)}{\leq} \mathbb{E}[F(\mathbf{w}^t) - \eta \cdot \nabla F(\mathbf{w}^t)^T \mathbf{G}^t + \frac{L}{2}\eta^2 ||\mathbf{G}^t||^2 \mid \mathbf{w}^t]$$
$$= F(\mathbf{w}^t) - \eta \cdot \mathbb{E}[\nabla F(\mathbf{w}^t)^T \mathbf{G}^t \mid \mathbf{w}^t] + \frac{\eta^2 L}{2}\mathbb{E}[||\mathbf{G}^t||^2 \mid \mathbf{w}^t], \tag{6}$$

where (a) is derived based on Assumption (5).

Firstly, we estimate the value of $\mathbb{E}[\nabla F(\mathbf{w}^t)^T \mathbf{G}^t \mid \mathbf{w}^t]$.

Since there are at most $r$ Byzantine workers, at most $r$ buffers may contain Byzantine gradients. Without loss of generality, suppose only the first $r$ buffers may contain Byzantine gradients.

Let $\mathbf{G}_{syn}^t = Aggr(\mathbf{h}_1, \ldots, \mathbf{h}_r, \mathbf{h}_{r+1}', \ldots, \mathbf{h}_B')$, where $\mathbf{h}_1, \ldots, \mathbf{h}_r$ may contain Byzantine gradients and be arbitrary value, and $\mathbf{h}_{r+1}', \ldots, \mathbf{h}_B'$ each stores loyal gradients computed based on $\mathbf{w}^t$. Thus,

$$\mathbb{E}[\nabla F(\mathbf{w}^t)^T \mathbf{G}_{syn}^t \mid \mathbf{w}^t] \geq ||\nabla F(\mathbf{w}^t)||^2 - A_1, \tag{7}$$

$$\mathbb{E}[||\mathbf{G}_{syn}^t||^2 \mid \mathbf{w}^t] \leq (A_2)^2. \tag{8}$$

Let $\alpha = 2\eta^2 L^2 \tau_{max}^2 (B - r) < 1$.

We claim that

$$\mathbb{E}[||\mathbf{G}^t - \mathbf{G}_{syn}^t||^2 \mid \mathbf{w}^t] \leq \left(\frac{1}{2}\alpha^{t+1} + \frac{\alpha}{1-\alpha}\right) \cdot (A_2)^2,$$

and

$$\mathbb{E}[||\mathbf{G}^t||^2 \mid \mathbf{w}^t] \leq \left(\alpha^{t+1} + \frac{2}{1-\alpha}\right) \cdot (A_2)^2.$$

Now we prove it by induction on $t$.

Step 1. When $t = 0$, all gradients are computed according to $\mathbf{w}^0$, and we have $\mathbf{G}^0 = \mathbf{G}_{syn}^0$. Thus,

$$\mathbb{E}[||\mathbf{G}^0 - \mathbf{G}_{syn}^0||^2 \mid \mathbf{w}^0] = 0 \leq \left(\frac{1}{2}\alpha^1 + \frac{\alpha}{1-\alpha}\right) \cdot (A_2)^2,$$

$$\mathbb{E}[||\mathbf{G}^0||^2 \mid \mathbf{w}^0] = \mathbb{E}[||\mathbf{G}_{syn}^0||^2 \mid \mathbf{w}^0] \leq (A_2)^2 \leq \left(\alpha^1 + \frac{2}{1-\alpha}\right) \cdot (A_2)^2.$$

Step 2. If

$$\mathbb{E}[||\mathbf{G}^{t'} - \mathbf{G}_{syn}^{t'}||^2 \mid \mathbf{w}^{t'}] \leq \left(\frac{1}{2}\alpha^{t'+1} + \frac{\alpha}{1-\alpha}\right) \cdot (A_2)^2,$$

$$\mathbb{E}[||\mathbf{G}^{t'}||^2 \mid \mathbf{w}^{t'}] \leq \left(\alpha^{t'+1} + \frac{2}{1-\alpha}\right) \cdot (A_2)^2,$$

holds for all $t' = 0, 1, \ldots, t - 1$ (induction hypothesis), then:

$$\mathbb{E}[||\mathbf{G}^t - \mathbf{G}_{syn}^t||^2 \mid \mathbf{w}^t]$$
$$= \mathbb{E}[||Aggr(\mathbf{h}_1, \ldots, \mathbf{h}_r, \mathbf{h}_{r+1}, \ldots, \mathbf{h}_B) - Aggr(\mathbf{h}_1, \ldots, \mathbf{h}_r, \mathbf{h}_{r+1}', \ldots, \mathbf{h}_B')||^2 \mid \mathbf{w}^t]$$
$$\overset{(b)}{\leq} \mathbb{E}[\sum_{b=r+1}^{B} ||\mathbf{h}_b - \mathbf{h}_b'||^2 \mid \mathbf{w}^t]$$

$$= \sum_{b=r+1}^{B} \mathbb{E}[\|\frac{1}{N_b^t} \sum_{i=1}^{N_b^t} (\nabla f(\mathbf{w}^{t_k}; z_{i_k}) - \nabla f(\mathbf{w}^t; z_{i_k}))\|^2 \mid \mathbf{w}^t]$$

$$\overset{(c)}{\leq} \sum_{b=r+1}^{B} \mathbb{E}[\frac{1}{N_b^t} \sum_{i=1}^{N_b^t} \|\nabla f(\mathbf{w}^{t_k}; z_{i_k}) - \nabla f(\mathbf{w}^t; z_{i_k})\|^2 \mid \mathbf{w}^t]$$

$$\overset{(d)}{\leq} \sum_{b=r+1}^{B} \mathbb{E}[\frac{1}{N_b^t} \sum_{i=1}^{N_b^t} L^2 \|\mathbf{w}^{t_k} - \mathbf{w}^t\|^2 \mid \mathbf{w}^t]$$

$$= \frac{L^2(B-r)}{N_b^t} \sum_{i=1}^{N_b^t} \mathbb{E}[\|\mathbf{w}^{t_k} - \mathbf{w}^t\|^2 \mid \mathbf{w}^t]$$

$$= \frac{L^2(B-r)}{N_b^t} \sum_{i=1}^{N_b^t} \mathbb{E}[\|\sum_{t'=t_k}^{t-1} \eta \cdot \mathbf{G}^{t'}\|^2 \mid \mathbf{w}^t]$$

$$\overset{(e)}{\leq} \frac{\eta^2 L^2(B-r)}{N_b^t} \sum_{i=1}^{N_b^t} \mathbb{E}[(t-t_k) \sum_{t'=t_k}^{t-1} \|\mathbf{G}^{t'}\|^2 \mid \mathbf{w}^t]$$

$$\overset{(f)}{\leq} \frac{\eta^2 L^2(B-r)}{N_b^t} \sum_{i=1}^{N_b^t} [(t-t_k) \sum_{t'=t_k}^{t-1} (\alpha^{t'+1} + \frac{2}{1-\alpha}) \cdot (A_2)^2]$$

$$\leq \frac{\eta^2 L^2(B-r)}{N_b^t} \sum_{i=1}^{N_b^t} [(t-t_k) \sum_{t'=t_k}^{t-1} (\alpha^t + \frac{2}{1-\alpha}) \cdot (A_2)^2]$$

$$\overset{(g)}{\leq} (\eta^2 L^2(B-r)\tau_{max}^2) \cdot (\alpha^t + \frac{2}{1-\alpha}) \cdot (A_2)^2$$

$$\overset{(h)}{\leq} \frac{1}{2}\alpha \cdot (\alpha^t + \frac{2}{1-\alpha}) \cdot (A_2)^2$$

$$= (\frac{1}{2}\alpha^{t+1} + \frac{\alpha}{1-\alpha}) \cdot (A_2)^2, \tag{9}$$

where (b) is derived based on the definition of stable aggregation function, (c) is derived based on Cauchy's Inequality, (d) is derived based on Assumption 5, (e) is also derived based on Cauchy's Inequality, (f) is derived based on induction hypothesis, (g) is derived based on that $t - t_k \leq \tau_{max}$, and (h) is derived based on that $\alpha = 2\eta^2 L^2 \tau_{max}^2 (B-r)$.

Therefore,

$$\mathbb{E}[\|\mathbf{G}^t\|^2 \mid \mathbf{w}^t] = \mathbb{E}[\|\mathbf{G}_{syn}^t + (\mathbf{G}^t - \mathbf{G}_{syn}^t)\|^2 \mid \mathbf{w}^t]$$

$$\overset{(i)}{\leq} 2 \cdot \mathbb{E}[\|\mathbf{G}_{syn}^t\|^2 \mid \mathbf{w}^t] + 2 \cdot \mathbb{E}[\|\mathbf{G}^t - \mathbf{G}_{syn}^t\|^2 \mid \mathbf{w}^t]$$

$$\overset{(j)}{\leq} 2 \cdot (A_2)^2 + 2 \cdot \mathbb{E}[\|\mathbf{G}^t - \mathbf{G}_{syn}^t\|^2 \mid \mathbf{w}^t]$$

$$\overset{(k)}{\leq} 2 \cdot (A_2)^2 + 2 \cdot (\frac{1}{2}\alpha^{t+1} + \frac{\alpha}{1-\alpha}) \cdot (A_2)^2$$

$$= (\alpha^{t+1} + \frac{2}{1-\alpha}) \cdot (A_2)^2, \tag{10}$$

where (i) is derived based on that $\|\mathbf{x} + \mathbf{y}\|^2 \leq 2\|\mathbf{x}\|^2 + 2\|\mathbf{y}\|^2$, $\forall \mathbf{x}, \mathbf{y} \in \mathbb{R}^d$, (j) is derived by the definition of $(A_1, A_2)$-effective aggregation function, and (k) is derived based on Inequality (9).

By Inequality (9) and (10), the claimed property also holds for $t' = t$.

In conclusion, for all $t = 0, 1, \ldots, T - 1$, we have:

$$\mathbb{E}[\|\mathbf{G}^t - \mathbf{G}_{syn}^t\|^2 \mid \mathbf{w}^t] \leq (\frac{1}{2}\alpha^{t+1} + \frac{\alpha}{1-\alpha}) \cdot (A_2)^2, \tag{11}$$

and

$$\mathbb{E}[\|\mathbf{G}^t\|^2 \mid \mathbf{w}^t] \le (\alpha^{t+1} + \frac{2}{1-\alpha}) \cdot (A_2)^2. \tag{12}$$

Also, $\mathbb{E}[\|\mathbf{G}^t\| \mid \mathbf{w}^t]^2 + Var[\|\mathbf{G}^t\| \mid \mathbf{w}^t] = \mathbb{E}[\|\mathbf{G}^t\|^2 \mid \mathbf{w}^t]$. Therefore,

$$\mathbb{E}[\|\mathbf{G}^t\| \mid \mathbf{w}^t] = \sqrt{\mathbb{E}[\|\mathbf{G}^t\| \mid \mathbf{w}^t]^2} \le \sqrt{\alpha^{t+1} + \frac{2}{1-\alpha}} \cdot A_2. \tag{13}$$

We have:

$$
\begin{aligned}
&\eta \cdot \mathbb{E}[\nabla F(\mathbf{w}^t)^T \mathbf{G}^t \mid \mathbf{w}^t] \\
=&\eta \cdot \mathbb{E}[\nabla F(\mathbf{w}^t)^T \mathbf{G}_{syn}^t \mid \mathbf{w}^t] + \eta \cdot \mathbb{E}[\nabla F(\mathbf{w}^t)^T (\mathbf{G}^t - \mathbf{G}_{syn}^t) \mid \mathbf{w}^t] \\
\overset{(l)}{\ge}&\eta \cdot (\|\nabla F(\mathbf{w}^t)\|^2 - A_1) + \eta \cdot \mathbb{E}[\nabla F(\mathbf{w}^t)^T (\mathbf{G}^t - \mathbf{G}_{syn}^t) \mid \mathbf{w}^t] \\
\ge&\eta \cdot \|\nabla F(\mathbf{w}^t)\|^2 - \eta \cdot A_1 - \eta \cdot \|\nabla F(\mathbf{w}^t)\| \cdot \|\mathbb{E}[(\mathbf{G}^t - \mathbf{G}_{syn}^t) \mid \mathbf{w}^t]\| \\
\overset{(m)}{\ge}&\eta \cdot \|\nabla F(\mathbf{w}^t)\|^2 - \eta \cdot A_1 - \eta \cdot D \cdot \|\mathbb{E}[(\mathbf{G}^t - \mathbf{G}_{syn}^t) \mid \mathbf{w}^t]\| \\
\overset{(n)}{\ge}&\eta \cdot \|\nabla F(\mathbf{w}^t)\|^2 - \eta \cdot A_1 - \eta \cdot D \cdot \sqrt{\frac{1}{2}\alpha^{t+1} + \frac{\alpha}{1-\alpha}} \cdot A_2, \tag{14}
\end{aligned}
$$

where (l) is derived based on the definition of $(A_1, A_2)$-effective aggregation function, (m) is derived by Assumption 3, and (n) is derived based on Inequality (11).

Combining Inequalities (6), (12), (14) and taking total expectation, we have:

$$
\begin{aligned}
\mathbb{E}[F(\mathbf{w}^{t+1})] \le& \mathbb{E}[F(\mathbf{w}^t)] - \eta \cdot \mathbb{E}[\|\nabla F(\mathbf{w}^t)\|^2] \\
&+ \eta \cdot A_1 + \eta \cdot D\sqrt{\frac{1}{2}\alpha^{t+1} + \frac{\alpha}{1-\alpha}} \cdot A_2 + \frac{1}{2}\eta^2 L(\alpha^{t+1} + \frac{2}{1-\alpha}) \cdot (A_2)^2.
\end{aligned}
$$

By telescoping, we have:

$$
\begin{aligned}
\eta \cdot \sum_{t=0}^{T-1} \mathbb{E}[\|\nabla F(\mathbf{w}^t)\|^2] \le& \{F(\mathbf{w}^0) - \mathbb{E}[F(\mathbf{w}^T)]\} + \frac{1}{2}\eta^2 TL(\alpha + \frac{2}{1-\alpha}) \cdot (A_2)^2 \\
&+ \eta T A_1 + \eta T D \cdot \sqrt{\frac{1}{2}\alpha + \frac{\alpha}{1-\alpha}} \cdot A_2.
\end{aligned}
$$

Divide both sides of the equation by $\eta T$, and let $\eta = O(\frac{1}{\sqrt{LT}})$:

$$
\begin{aligned}
&\frac{\sum_{t=0}^{T-1} \mathbb{E}[\|\nabla F(\mathbf{w}^t)\|^2]}{T} \\
\le&\frac{\{F(\mathbf{w}^0) - \mathbb{E}[F(\mathbf{w}^T)]\}}{\eta T} + \frac{1}{2}\eta L(\alpha + \frac{2}{1-\alpha}) \cdot (A_2)^2 + A_1 + D \cdot \sqrt{\frac{1}{2}\alpha + \frac{\alpha}{1-\alpha}} \cdot A_2 \\
\le&\frac{\sqrt{L}[F(\mathbf{w}^0) - F^*]}{\sqrt{T}} + \frac{\sqrt{L}(\frac{1}{2}\alpha + \frac{1}{1-\alpha}) \cdot (A_2)^2}{\sqrt{T}} + A_1 + \alpha^{\frac{1}{2}}[\frac{3-\alpha}{2(1-\alpha)}]^{\frac{1}{2}} \cdot DA_2.
\end{aligned}
$$

Note that $\alpha = 2\eta^2 L^2 \tau_{max}^2 (B-r) = O\left(\frac{L\tau_{max}^2(B-r)}{T}\right)$, finally we have:

$$
\begin{aligned}
\frac{\sum_{t=0}^{T-1} \mathbb{E}[\|\nabla F(\mathbf{w}^t)\|^2]}{T} \le& O\left(\frac{\sqrt{L} \cdot [F(\mathbf{w}^0) - F^*]}{\sqrt{T}}\right) + O\left(\frac{\sqrt{L}(A_2)^2(1+\alpha)}{\sqrt{T}}\right) \\
&+ O\left(\alpha^{\frac{1}{2}} DA_2\right) + A_1 \\
=& O\left(\frac{L^{\frac{1}{2}}[F(\mathbf{w}^0) - F^*]}{T^{\frac{1}{2}}}\right) + O\left(\frac{L^{\frac{1}{2}}\tau_{max}(B-r)^{\frac{1}{2}} DA_2}{T^{\frac{1}{2}}}\right)
\end{aligned}
$$

$$+ O\left(\frac{L^{\frac{1}{2}}(A_2)^2}{T^{\frac{1}{2}}}\right) + O\left(\frac{L^{\frac{5}{2}}(A_2)^2\tau_{max}^2(B-r)}{T^{\frac{3}{2}}}\right) + A_1.$$

Specailly, when $B = O(r)$, we have:

$$\frac{\sum_{t=0}^{T-1}\mathbb{E}[\|\nabla F(\mathbf{w}^t)\|^2]}{T} \leq O\left(\frac{L^{\frac{1}{2}}[F(\mathbf{w}^0) - F^*]}{T^{\frac{1}{2}}}\right) + O\left(\frac{L^{\frac{1}{2}}\tau_{max}DA_2r^{\frac{1}{2}}}{T^{\frac{1}{2}}}\right)$$
$$+ O\left(\frac{L^{\frac{1}{2}}(A_2)^2}{T^{\frac{1}{2}}}\right) + O\left(\frac{L^{\frac{5}{2}}(A_2)^2\tau_{max}^2r}{T^{\frac{3}{2}}}\right) + A_1.$$

$\square$

### B.6 PROOF OF PROPOSITION 2

*Proof.* Under the condition that $\forall \mathbf{w}^t \in \mathbb{R}^d$, $\mathbb{E}[\|\mathbf{G}_{syn}^t - \nabla F(\mathbf{w}^t)\| \leq D \mid \mathbf{w}^t]$, we have:

$$\mathbb{E}[\nabla F(\mathbf{w}^t)^T\mathbf{G}_{syn}^t \mid \mathbf{w}^t]$$
$$= \mathbb{E}[\nabla F(\mathbf{w}^t)^T \left[\nabla F(\mathbf{w}^t) + (\mathbf{G}_{syn}^t - \nabla F(\mathbf{w}^t)) \mid \mathbf{w}^t]\right.$$
$$= \|\nabla F(\mathbf{w}^t)\|^2 + \mathbb{E}[\nabla F(\mathbf{w}^t)^T(\mathbf{G}_{syn}^t - \nabla F(\mathbf{w}^t)) \mid \mathbf{w}^t]$$
$$\geq \|\nabla F(\mathbf{w}^t)\|^2 - \|\nabla F(\mathbf{w}^t)\| \cdot \mathbb{E}[\|\mathbf{G}_{syn}^t - \nabla F(\mathbf{w}^t)\| \mid \mathbf{w}^t]$$
$$\geq \|\nabla F(\mathbf{w}^t)\|^2 - D \times D$$
$$= \|\nabla F(\mathbf{w}^t)\|^2 - D^2.$$

Combining with the property (i) of $(A_1, A_2)$-effective aggregation function, we have $A_1 \leq D^2$.

$\square$

## C  MORE EXPERIMENTAL RESULTS

Figure 4, Figure 5 and Figure 6 illustrate the average training loss w.r.t. epochs when there are no Byzantine workers, 3 Byzantine workers and 6 Byzantine workers. Please note that in Figure 5 and Figure 6, some curves do not appear, because the value of loss function is extremely large or even exceeds the range of floating-point numbers, due to the Byzantine attack. $\gamma$ is the hyper-parameter about the assumed number of Byzantine workers in Kardam. The experimental results about training loss give further support to the experimental summary in Section 5.

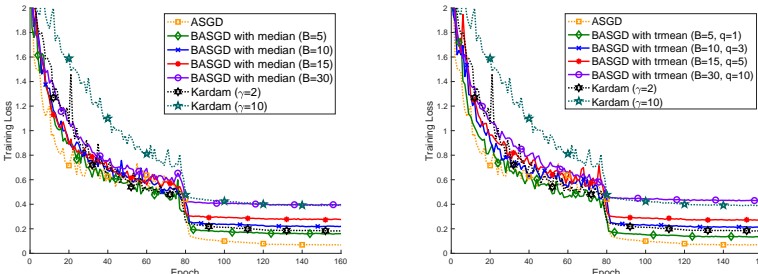

Figure 4: Average training loss w.r.t. epochs when there are no Byzantine workers. The aggregation function in BASGD is set to be median (left) and trimmed-mean (right), respectively.

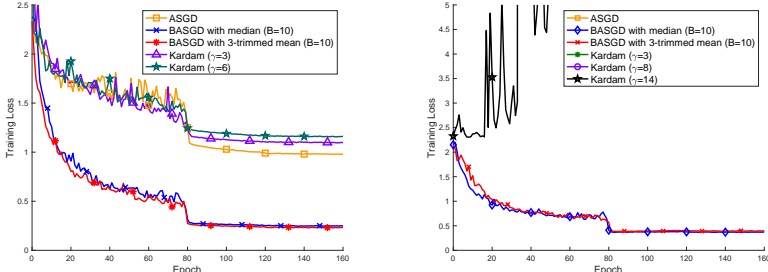

Figure 5: Average training loss w.r.t. epochs in face of random disturbance attack (left) and negative gradient attack (right), when the number of Byzantine workers $r = 3$. Some curves do not appear in the figure, because the value of loss function is extremely large or even exceeds the range of floating-point numbers.

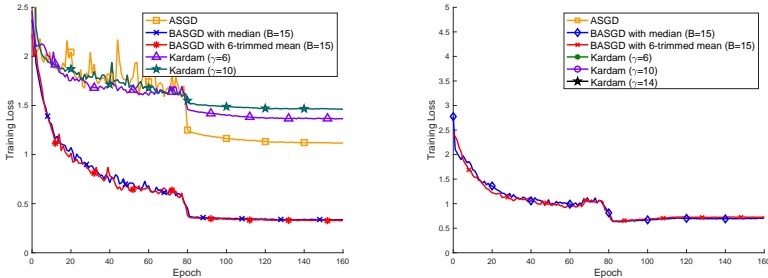

Figure 6: Average training loss w.r.t. epochs in face of random disturbance attack (left) and negative gradient attack (right), when the number of Byzantine workers $r = 6$. Some curves do not appear in the figure, because the value of loss function is extremely large or even exceeds the range of floating-point numbers.

