# OpenReview forum: "BASGD: Buffered Asynchronous SGD for Byzantine Learning"
_ICLR.cc/2021/Conference — Reject_

### Official Review · AnonReviewer3 · 2020-10-24
**Good paper, but not fully asynchronous.**

**Rating:** 5
**Confidence:** 4

**Review:**

Review: This paper proposes BASGD which uses buffers to perform asynchronous Byzantine learning. In each SGD step, all workers compute gradients and send them to the server where their ad buffer is updated. When all of the buffers are updated, the server performs an model update. When a worker send a gradient to the server, it also pulls the latest model and compute the gradient on it no matter the server update the model or not. The main contribution in this paper is to introduce a new approach to do asynchronous Byzantine learning without storing training samples on the server like zeno++.

=======================================================

Pros:

+ The problem of Byzantine learning in the asynchronous environment is interesting and has rarely been studied, especially under the assumption that the server has no training data.

+ The idea of using buffer on the server to achieve (partial) asynchrony is interesting.

+ This paper provides extensive experiments to demonstrate its effectiveness.

=======================================================

Concerns:

- BASGD is not fully asynchronous as the paper claims to be. The server only updates the model when all of the buffers are non-zero. That is, the whole system will be slowed down by the slowest buffer. The remaining good workers, at the time of waiting, are computing the gradients on the same model weight. In this case, BASGD resembles SBL with larger batch size but at the cost of tolerating less Byzantine workers.

- This paper does not explicitly state how to choose the number of buffers $B$ in order to achieve both asynchrony and Byzantine-robustness. In page 3, there are "BASGD introduces $B$ buffers ($0<B\le m$) on server". However, when $B$ is small, there is no robustness; when $B$ is large, there is no asynchrony.

- The Definition 1, 4, 5, are not common in Byzantine robust learning. Are these definitions used only to make the proof easier?

- The results in the theorems are not clearly presented. The $O(1/T)$ in Theorem 1 and 2 does not reflect how $B$, $q$ influence the convergence rate.

=======================================================

Minor comments:

- The negative gradient attack and random disturbance attack are very easy to defend. It would be better to choose some more challenging attacks.

- It is better to improve the writing in section 4 for better readability.

---

> ### Author Response · Authors · 2020-11-19
> **Reply R3: We modify the improper statement, and address the concerns.**
>
> We thank reviewer R3 for reading our work and their insightful suggestions. We address the concerns point by point as follows:
>
> ------
> **BASGD is not fully asynchronous as the paper claims to be. The whole system will be slowed down by the slowest buffer.**
> + We are sincerely appreciated to reviewer R3 for pointing out our improper statements. We have modified the statement in the latest rebuttal revised version.
> + Besides, we would like to point out that the asynchrony is greatly damaged only when the buffer number $B$ is close to $m$. It is true that the whole system will be slowed down by the slowest buffer. However, the probability of its occurrence is very small when $B$ is not that large. More specifically, there are about $(m/B)$ workers corresponds to each buffer. A buffer will become a straggler only when the $(m/B)$ workers are all stragglers. The probability is relatively low when $(m/B)$ is not too small.
> + In real-world federated learning settings, there may be hundreds of millions of mobile devices used as workers [1], and the Byzantine worker number $r$ will be much smaller than $m$. In this case, $B$ is suggested to be set large enough to have resilience against $\hat{r}$ (a relatively large estimation of $r$ in advance, according to practical application) Byzantine workers. Therefore, $(m/B)$ is not a small number. The probability is very low that $(m/B)$ workers are all stragglers.
> + In addition, even if the server's model weight parameter is not updated when receiving worker$_k$'s gradient, the parameter sent back to worker$_k$ may also be newer than worker$_k$'s. For example, worker$_k$ computed the gradient based on its local parameter $w_1$. During its computation, server's parameter has already been updated to $w_2$. When worker$_k$ sends gradient to the server, server may not execute an SGD step, but the parameter sent back (i.e., $w_2$) is newer than worker$_k$'s (i.e., $w_1$).
>
> ------
> **This paper does not explicitly state how to choose the number of buffers $B$ in order to achieve both asynchrony and Byzantine-robustness.**
> + We thank reviewer R3 for the suggestion. Discussion on how to choose $B$ is added at the end of Section 3 in the latest revised version.
>
> ------
> **The Definition 1, 4, 5, are not common in Byzantine robust learning. Are these definitions used only to make the proof easier?**
> + We will clarify that the definitions are ***not*** used to make the proof easier. If we wanted to make the proof easier, we could prove the convergence for a certain aggregation function (e.g., coordinate-wise median). Median satisfies both Definition 1 (Proposition 1), Definition 4 (not hard to check), and Definition 5 (have already been proved [2]). The convergence proof for median only will be much easier.
> + Since BASGD can be seen as a technique of asynchronization, we want to prove the convergence of BASGD not only for a distinct aggregation, but also a class of aggregation functions. However, to the best of our knowledge, there is still a lack of consensus on what properties a 'good' aggregation function should have. Due to the reasons above, we define these two classes of aggregation functions. The motivation is explained after the definition.
>
> ------
> **The $O(1/T)$ in Theorem 1 and 2 do not reflect how $B$, $q$ influence the convergence rate.**
> + We thank review R3 for the insightful comment. We present more details in Theorem 1 and Theorem 2 in the latest revised version.
> + Detailed discussion about $B$, $q$, and $r$ is after the proof of Proposition 2 in Appendix in the latest revised version.  We add the statement about $q$ and $r$ after Theorem 1.
> + The detailed discussion about how $B$ influences the rate involves more complex proof details. For space-saving and readability, we decided not to present it in the main text. The conclusion is that, the extra constant variance increases as $B$ increases. Larger $B$ brings more Byzantine resilience, while brings more variance as well. So, $B$ should be set properly, as added at the end of Section 3.
>
> ------
> **It is better to improve the writing in section 4 for better readability.**
> + In the latest revised version, we re-organize Section 4 by moving lengthy definitions to Appendix, and present more details of Theorem 1 and Theorem 2. Some extra discussion is also added. We hope that it could improve the readability of our work. Thanks again for the reviewer's suggestion. We hope that the reviewer could re-evaluate our work according to the revised version.
>
> ------
> References:
>
> [1]. Peter Kairouz, H Brendan McMahan, Brendan Avent, Aure ́lien Bellet, Mehdi Bennis, Arjun Nitin Bhagoji, Keith Bonawitz, Zachary Charles, et al. Advances and open problems in federated learning. arXiv:1912.04977, 2019.
>
> [2]. Dong Yin, Yudong Chen, Ramchandran Kannan, and Peter Bartlett. Byzantine-robust distributed learning: Towards optimal statistical rates. In Proceedings of the International Conference on Machine Learning, pp. 5650–5659, 2018.

---

### Official Review · AnonReviewer2 · 2020-10-27
**Ok but not good enough**

**Rating:** 5
**Confidence:** 3

**Review:**

The paper proposes a practical asynchronous stochastic gradient descent for Byzantine distributed learning where some of transmitted gradients are likely to be replaced by arbitrary vectors. Specifically, the server temporarily stores gradients on multiple  (namely $B$) buffers and performs a proper robust aggregation to compute a more robust from them. When $B = 1$,  BASGD is reduced to ASGD. They also conduct experiments to show the performance of BASGD.

The paper is well written and easy to follow. All proof seems correct though I didn’t check very carefully. However, there are some issues:
1. Compared to other asynchronous SGD methods, one advantage of BASGD is it doesn’t have the need of storing any samples on the server. However, I didn’t understand why such property is meaningful. The authors declare that it helps BASGD take less risk of privacy leakage. Noting the server store gradients on buffers and gradients may leak privacy [1]. If a third party can have access to the buffers, privacy leak can still happen. BASGD didn’t use techniques like differential privacy, so it seems ill-founded to say “BASGD takes less risk of privacy leakage”.
2. In my opinion, Theorem 1 doesn’t guarantee that BASGD is able to find a stationary point of $F$ since the extra constant variance will not vanish when $T$ goes infinity. By contrast, ZENO++, a robust fully asynchronous SGD, could ensure convergence to a stationary point. This strikes me that the theorem is quite weak. Theorem 2 has the same problem.
3. It makes me feel strange to assume the gradient is biased (Assumption 2). For example, If we want to minimize $F(w)$ but we use stochastic gradients computed from another $\tilde{F}(w)$ (that is totally different from $F(w)$), could we still guarantee the algorithm converges to the stationary point of $F(w)$? I am afraid the answer is NO. This thought experiment not only shows using biased gradients doesn’t help convergence but also shows Theorem 1 doesn’t guarantee the convergence of BASGD.
4. Some parts of the algorithm are not well explored. For example, how the performance of BASGD changes when we vary the value of $q$ or the value of $B$. The experiment didn’t explore these aspects. Besides, Noting that the number of buffers $B$ is quite important for good performance, however, there is no investigation on how $B$ affects convergence and no suggestion on how to choose a proper $B$. In experiments, $B$ is set in advance for no reason.

[1] Ligeng Zhu, Zhijian Liu, and Song Han. Deep leakage from gradients. In Advances in Neural Information Processing Systems, pages 14747–14756, 2019.

---------
I have read the authors' response. The authors have addressed most of my concerns, but I still think the motivation is a little farfetched. Considering the paper indeed explorees some aspects (in theories and experiments) of the use of buffers in asynchronous Byzantine Learning, I will improve my point to 5.

---

> ### Author Response · Authors · 2020-11-17
> **Reply R2: We have addressed all the raised concerns**
>
> We greatly thank the reviewer for their summary and comments. We believe that we have addressed all the raised concerns as follows. We will appreciate it if the reviewer could re-evaluate our work.
>
> ------
> **One advantage of BASGD is it doesn’t have the need of storing any samples on the server. However, I didn’t understand why such property is meaningful.**
> + In traditional machine learning, it is OK to store samples on the server. However, storing samples on the server may be no longer possible in federated learning (FL). In the setting of FL, data should be keeping decentralized to mitigate many of the systemic privacy risks [1]. Therefore, it is difficult for the service provider (server) to store a large number of high-quality instances.
> + The reviewer said that storing gradients may also leak privacy. However, we would like to point out that it is still an open problem whether transmitting gradients will cause privacy leakage. There are also many recent works presenting or improving differential privacy [2], which is a technique for privacy defense. Besides,  homomorphic encryption is also a powerful privacy-defense technique [3].
> + In conclusion, storing samples is not possible in the setting of federated learning, where workers are no longer under the control of service provider. In this case, Byzantine attacker is more likely to appear. Thus, we believe the property that BASGD doesn't need storing any samples on the server is meaningful.
>
> ------
> **Theorem 1 doesn’t guarantee that BASGD is able to find a stationary point of $F$, while Zeno++ could ensure convergence to a stationary point.**
> + We have to point out that the convergence of Zeno++ [4] is under an assumption called Polyak-Łojasiewicz (PL) inequality, which says $\exists \mu >0,$ such that $\forall x,\ f(x)-f(x_*)\leq\frac{1}{2\mu}||\nabla f(x)||^2$. This is a very strong assumption, under which there is only one stationary point (global minimum $x_*$). In addition, the norm of gradient $||\nabla f(x)||^2$ is bounded below by $2\mu[f(x)-f(x_*)]$. Many machine learning models do not satisfy this property. Besides, Zeno can be seen as the synchronous version of Zeno++. The convergence proof of Zeno is under the general non-convex assumption, and there is also an extra constant term [5].
> Many Byzantine learning methods use robust aggregation to obtain the gradient for updating. However, there is a gap between the aggregated result and the true global gradient $\nabla F(w_t)$, because of the Byzantine attack. A biased gradient is used for parameter updating. Under this circumstance, it is hardly possible for the method to be guaranteed to converge to a stationary point.
>
> ------
> **It makes me feel strange to assume the gradient is biased (Assumption 2). Using biased gradients doesn’t help convergence.**
> + The biased gradient assumption is from the problem setting of federated learning, where the instances stored on different workers are non-IID. Each worker can only obtain a biased estimation of global gradient $\nabla F(w_t)$ based on its stored instances and the parameter $w_t$. As the review said, it is more difficult to guarantee the convergence when the gradient is biased, let alone there are Byzantine workers. Thus, we believe the theoretical result is acceptable that $\frac{1}{T}\sum_{t=0}^{T-1}\mathbb{E}[||\nabla F(w_t)||]^2$ is bounded by a relative small positive value.
>
> ------
> **The influence of buffer number $B$ is not well explored.  Besides, there is no investigation on how $B$ affects convergence and no suggestion on how to choose a proper $B$.**
> + The effect of $B$ is empirically explored in the image classification experiments. The experimental results are presented in Figure 2(a) and Figure 2(d). Many thanks for your constructive suggestion. We add the discussion of the effect of $B$ and give suggestions for the choice of $B$ at the end of Section 3 in the rebuttal revised version. We will appreciate it if the reviewer could re-evaluate our work.
>
> ------
>
> [1]. Peter Kairouz, H Brendan McMahan, Brendan Avent, Aure ́lien Bellet, Mehdi Bennis, Arjun Nitin Bhagoji, Keith Bonawitz, Zachary Charles, Graham Cormode, Rachel Cummings, et al. Advances and open problems in federated learning. arXiv:1912.04977, 2019.
>
> [2]. Clément L Canonne, Gautam Kamath, Thomas Steinke. The Discrete Gaussian for Differential Privacy. In Advances in Neural Information Processing Systems, 2020.
>
> [3]. Zhang C, Li S, Xia J, et al. Batchcrypt: Efficient homomorphic encryption for cross-silo federated learning. In Proceedings of the USENIX Annual Technical Conference, 2020.
>
> [4]. Cong Xie, Sanmi Koyejo, and Indranil Gupta. Zeno++: Robust fully asynchronous SGD. In Proceedings of the International Conference on Machine Learning, 2020.
>
> [5]. Cong Xie, Sanmi Koyejo, and Indranil Gupta. Zeno: Distributed stochastic gradient descent with suspicion-based fault-tolerance. In Proceedings of the International Conference on Machine Learning, pp. 6893–6901, 2019.

---

### Official Review · AnonReviewer1 · 2020-10-28
**Interesting method**

**Rating:** 6
**Confidence:** 3

**Review:**

#### Summary

This work proposes a method for Byzantine learning in a parameter-server setting using asynchronous updates, and without the need for storing training instances on the master node. They provide theoretical guarantees and some experiments on small-scale tasks.

---

#### Originality

Synchronous methods for byzantine learning enable one to compare communicated gradients with one another in order to filter out byzantine workers. When updates are asynchronous, such a procedure cannot be employed because gradients are not necessarily communicated to the master node at the same time. Previous work on Asynchronous byzantine learning stores training instances on the master node to alleviate this issue. This work proposes instead to use gradient buffers on the server to eliminate the need to store training instances on the master node; this is the main selling point of the algorithm, and deems the method sufficiently novel in my opinion.

However, the motivation for avoiding storing instances on the master node is not well fleshed out in my opinion. Though the applications to Federated Learning are mentioned, updates are typically performed locally on-device and *parameters* are communicated back to the master, and therefore this method is not applicable to that setting. It is unclear exactly what the privacy concerns are, but I would suggest reworking the motivation exposition a little.

---

#### Significance and Quality

The method is actually quite nice and intuitive. The significance relates back to improving the motivation, but the method may be of sufficient interest to the community.

*On the theory*:

Note that the bounded gradient assumption is very strong! when combined with the L-smoothness assumption, it implies a convergent subsequence a priori! In short, it is like assuming ahead of time that the algorithm convergence.

$N^t_b$ is used in the main paper (e.g., page 5), but is only defined in the appendix on page 14. I would suggest including a one sentence description that these are the number of gradients stored in buffer $b$ at iteration $t$.

Under asynchrony or non-iid data, theorem 1 does not guarantee convergence to a stationary point… even with a diminishing step-size. Moreover, even with iid data and fully synchronicity and a diminishing step-size, the algorithm still does not converge to a stationary point due to the extra variance term on the r.h.s. I do not mean to criticize the results; only to point out this fact. Moreover, these bounds are somewhat vacuous due to the presence of the non-degrading $D$ gradient boundedness term on the r.h.s of Theorem 1 in the variance term. With L-smoothness and the assumption of $D$-bounded gradients, as I mentioned above, all iterates of any objective (regardless of the algorithm), will remain with a ball of a stationary point, the size of which is proportional to $D$. (To the authors credit, I have seen similar bounds in previous byzantine learning methods).

For Theorem 2, as I understand it, the $\alpha^{1/2}$ term actually has a $\mathcal{O}(1/\sqrt{T})$ dependence, so unless i’m missing something, why not remove the constant $\alpha$ and substitute in a quantity that decays with $1/\sqrt{T}$, and state the theorem results for all $T \geq$ some threshold (to satisfy the current $\alpha < 1$ constraint). this reformulation will make it clearer that the only lingering (non-decaying factor) is $A_1$, which is due to gradient bias.

More generally, I am curious if there is a way to correct for the convergence errors and improve the results to guarantee convergence to a stationary point (and not some neighborhood thereof), given that a $\frac{1}{\sqrt{T}}$ step-size is employed.

---

#### Clarity

Work is sufficiently clear. One minor point is that Asynchronous SGD is missing a reference (common references for this method include Dean et al., or Bengio et al.)

---

> ### Author Response · Authors · 2020-11-23
> **Reply R1: We improve the theoretical result in the latest version. (1/2)**
>
> We greatly thank the reviewer for the concise summary of our work and the precious comments. Especially, we are deeply grateful for the insightful and constructive suggestions on improving the theoretical results. Besides, we sincerely apologize for not replying in time, because we thought that the reviewer revealed some really important points in the theory of our works. We have been trying to address the raised concerns, and feel really sorry for the late reply again.
>
> We have improved the results in the latest version according to the comments. The raised concerns are addressed point by point as follows. We will appreciate it if the reviewer could re-evaluate our work in light of the improvement.
>
> ------
> **The motivation for avoiding storing instances on the master node is not well fleshed out in my opinion.**
> + Sending gradients (differences) and sending parameters have close relations. In local updating methods, workers can also send the differences between the updated parameter and the original one. The difference may have more small elements (close to $0$), and can be compressed with less error. For example, Deep Gradient Compression [1] takes this way. Thus, sending gradients and sending parameters may not have an explicit border. Besides, BASGD may be used together with some other techniques (e.g., local updating) that solving some other problems in federated learning, although it needs further work to check. Therefore, we believe that this work may contribute to the knowledge of federated learning.
> + Then we explain the concerns about privacy. In the setting of FL, data should be keeping decentralized to mitigate many of the systemic privacy risks [2]. Therefore, it is difficult for the service provider (server) to store a large number of high-quality instances.
> Besides, in some practical applications, collecting users' data on the server may face the risk of privacy leakage (e.g., medical data or financial data). In these cases, collecting data is prohibited by privacy policy.
>
> ------
> **$N_b^t$ is used in the main paper (e.g., page 5), but is only defined in the appendix on page 14.**
> + Actually, $N_b^t$ is defined in Section 3.1 in the original version. We have found that the definition may be inconspicuous, and explained $N_b^t$ again in Section 4, for more readability.
>
> ------
> **Why not remove the constant $\alpha$ and substitute in a quantity that decays with $\frac{1}{\sqrt{T}}$?**
> + Much appreciation for the reviewer's insightful comment. Benefiting from it, the theoretical results get improved in the latest revised version. The only remaining constant term is $A_1$ now, exactly as the reviewer said.
>
> ------
> **Note that the bounded gradient assumption is very strong! When combined with the L-smoothness assumption, it implies a convergent subsequence a priori!**
> + We are grateful for the insightful comment, which reveals an important drawback of our work. As a complement, we add Proposition 2 after Theorem 2. The added proposition reveals that the squared norm of the gradient can decrease to $A_1$, which is smaller than $D^2$.
>
> ------
> **One minor point is that Asynchronous SGD is missing a reference.**
> + We have added the recommended reference (in the third paragraph of Section 2.1) in the latest revised version.
>
> ------
> References:
>
> [1]. Lin, Y., Han, S., Mao, H., Wang, Y., & Dally, B. (2018, February). Deep Gradient Compression: Reducing the Communication Bandwidth for Distributed Training. In International Conference on Learning Representations.
>
> [2]. Peter Kairouz, H Brendan McMahan, Brendan Avent, Aure ́lien Bellet, Mehdi Bennis, Arjun Nitin Bhagoji, Keith Bonawitz, Zachary Charles, Graham Cormode, Rachel Cummings, et al. Advances and open problems in federated learning. arXiv:1912.04977, 2019.

---

> ### Author Response · Authors · 2020-11-24
> **Reply R1: We improve the theoretical result in the latest version (2/2)**
>
> Due to the characters limit in a single reply, we separate our reply into two parts.
>
> ------
> **Under asynchrony or non-iid data, theorem 1 does not guarantee convergence to a stationary point. More generally, I am curious if there is a way to correct for the convergence errors and improve the results to guarantee convergence to a stationary point (and not some neighborhood thereof).**
> + It is true that each theorem does not guarantee convergence to a stationary point if there exists a non-zero constant term. Moreover, the theorems do not guarantee convergence to some neighborhood of a stationary point, as well. A counter-example is that $f(x)=0.01\cos (x)$. Even if we can guarantee $|f'(x^t)|\leq 0.01$, $x^t$ may locate on everywhere of $\mathbb{R}$, and move among different local minima.
> + Actually, we have been trying to improve the results. As indicated by Theorem 2, the main problem is the gradient bias. We will use a toy example to show this. Consider a one-dimensional optimization problem on a single machine. Objective function $f(x)=\frac{1}{2}x^2$, and $f'(x)=x$. However, some certain optimization method uses biased gradient $\hat{f}'(x)=x-1$ to update $x^t$. Then whatever the step size is, $x^t$ may only converge to $1$, instead of the true answer $0$. Therefore, even if the objective function is strongly convex, $L$-Lipschitz, and there is no variance and no Byzantine workers, a method using biased gradient can hardly be guaranteed to converge to global minimum.
> + There are at least three causes of bias in asynchronous Byzantine learning. (i) The first cause is the delay, while the bias caused by delay may decrease to $0$ as $T$ increases, under certain conditions. (ii) The second cause is the Byzantine workers. Bias caused by Byzantine workers is hard to be reduced to $0$ in general cases. The bias in Zeno++ is proved to be reduced to $0$ under the condition of PL-inequality. However, in general non-convex cases, due to the arbitrary performance of Byzantine workers, how to decrease gradient bias is still a tough problem. (iii) The third cause is the non-IID training instances across workers. Most existing robust estimator of mean is biased when training instances are non-IID.

---

### Official Review · AnonReviewer4 · 2020-11-03
**A decent paper, but a number of important points are not carefully discussed in it.**

**Rating:** 7
**Confidence:** 3

**Review:**

This paper studies distributed learning in the presence of Byzantine workers in the asynchronous setting. Its main contributions include generalization of the existing literature on Byzantine fault tolerance in distributed learning to incorporate the case of asynchronous learning. This generalization involves an algorithm, convergence analysis for the algorithm, and experimental results. While the results presented in the paper appear to be correct, I would like the authors to focus on the following points during their revision.

1. In Section 2.2, the definition of Byzantine worker, it is not clear why the worker is being indexed with $k_t$? What is the meaning of $t$ in this usage of the worker index?
2. The writing of the paper could use some proofreading. Some of the sentences are hard to parse on first read, while some other sentences suffer from grammatical errors. As a specific example, I could not understand the meaning of "Only when all buffers have got changed since the last SGD step, ..." in Section 3.1 until I reread the main parts of the paper.
3. Related to the previous point, the discussion in Section 3.1 in general is hard to parse because of the notation and could benefit from revision. Also, $m$ in this section is undefined up to this point in time and it is not clear what it means.
4. A number of aggregation functions have been proposed in prior works (see e.g. Adversary-resilient distributed and decentralized statistical inference and machine learning: An overview of recent advances under the Byzantine threat model). Do all of these previous aggregation functions satisfy the characterization of Section 3.2? It would be helpful to have some discussion of this.
5. Theorem 1 and Theorem 2 leave something to be desired. Since the task is to engage in distributed learning, one expects to see some sort of speedup from the fact that $n$ workers are being used to divide up the work. However this speed-up does not seem to be coming up in the analysis or the discussion. In the absence of such a speed-up, it is not clear if the authors are really providing guarantees that are useful for distributed learning.
6. It would be useful to discuss the impact of heavily delayed workers on the algorithm. What if the sum of the number of heavily delayed workers and Byzantine workers exceeds $r$?
7. The plots corresponding to the experimental results are too small and should be modified to have bigger font and size.

***Post-discussion period comments***
The authors have done an adequate job of responding to my queries and have also revised the paper in light of the comments of all the reviewers. While the paper could always be improved, I believe it is now above the threshold of acceptance and it should be accepted into the program, if possible. I am raising my score for this paper in light of the discussion and the revised paper.

---

> ### Author Response · Authors · 2020-11-22
> **Reply R4: We have revised our work in the latest version and addressed the raised concerns.**
>
> We greatly thank the reviewer R4 for the concise summary of our work and the insightful comments. Benefiting from the constructive suggestions, we have revised some improper or confusing statements. The raised concerns are addressed point by point as follows. We will greatly appreciate it if the reviewer could re-evaluate our work.
>
> ------
> **1. In Section 2.2, the definition of Byzantine worker, it is not clear why the worker is being indexed with $k_t$.**
> + We greatly thank the reviewer for pointing out our improper use of the index, and sincerely apologize for the confusion caused by this mistake. The index $t$ has been removed in the latest revised version.
>
> ------
> **2. I could not understand the meaning of "Only when all buffers have got changed since the last SGD step, ..." in Section 3.1 until I reread the main parts of the paper.**
> + We have changed the confusing sentence to "Only when each buffer has stored at least one gradient, ...", and revised some other statements that may cause confusion. We thank the reviewer for helping us to improve the readability.
>
> ------
> **3. In section 3.1, $m$ is undefined up to this point in time and it is not clear what it means.**
> + Actually, $m$ has been defined in the second paragraph of section 2.1, where it says, "Training instances are disjointedly distributed across $m$ workers", in the original version. Though, for more readability, we have mentioned the meaning of $m$ again in section 3.1, in the latest revised version.
>
> ------
> **4. A number of aggregation functions have been proposed in prior works. Do all of these previous aggregation functions satisfy the characterization of Section 3.2?**
>
> We have read the referenced work. We briefly list the checking results as follows:
> + As mentioned after Definition 5 in the original version, **Krum**, **multi-Krum**, **Coordinate-wise median**, and **Coordinate-wise trimmed mean** all satisfy the characterization.
> + **GeoMed** also satisfies both Definition 4 and Definition 5. However, the thorough proof is lengthy. For space-saving, we provide an intuitive idea here. The sum of $l_2$-norm form $\sum||y-y_i||$ reveals that the objective function is $1$-Lipschitz w.r.t. each $y_i$ and convex w.r.t. $y$. Thus, minima $y^*$ is $1$-Lipschitz w.r.t. each $y_i$, revealing that GeoMed satisfies Definition 4. Also, the aggregated result of GeoMed can be bounded by the combination of loyal gradients, since loyal gradients make up the majority. the two inequalities in Definition 5 can be proved by using this property.
> + **Bulyan** uses an existing aggregation rule to obtain a new one. Its performance is highly dependent on the chosen aggregation rule, and the property of Bulyan is difficult to be analyzed alone.
> + **SignSGD** uses quantization to compress gradients to only 1 bit (i.e. $+1$ or $-1$). Each worker votes for $+1$ or $-1$ on each coordinate, and the server uses the majority for updating. The voting resulting is equivalent to the coordinate-wise median of $+1$'s and $-1$'s. Thus, SignSGD can be seen as a combination of 1-bit quantization and coordinate-wise median aggregation. Coordinate-wise median satisfies the characterization, as discussed above.
> + **Zeno/Zeno++** Zeno++ itself is an asynchronous version of Zeno. It is meaningless to check for Zeno/Zeno++.
>
> ------
> **5. In the absence of such a speed-up, it is not clear if the authors are really providing guarantees that are useful for distributed learning.**
> + Firstly, we would like to point out that, in the setting of BASGD, training data or instances are distributedly stored on workers. The main purpose of BASGD is to make it possible to jointly training a model asynchronously without sending training data, even if there are Byzantine workers. To the best of our knowledge, BASGD is the first method to address the problem in this settings. This is the main contribution of our work.
> + In this problem setting, instances are locally stored on each worker, and gradients can only be locally computed by each worker. Thus, computation is distributed to each worker naturally by the problem setting. In cases without Byzantine workers, the computation of each worker is $O(1/m)$. Although speed-up is not the main contribution, we finally decide to add a brief discussion about complexity at the end of section 3, in order to make the work more complete.
>
> ------
> **6.What if the sum of the number of heavily delayed workers and Byzantine workers exceeds $r$?**
> + In some existing works, heavy delay is seen as a kind of Byzantine failure [1]. Stale gradients may provide no help for convergence. Though not purposely adversarial, heavily delayed workers may damage the convergence. Thus, convergence is hard to guarantee in this case, though the method may accidentally succeed.
>
> ------
> **7. Plots should have bigger font and size.**
> + We have modified it in the latest version.
>
> [1]. Peter Kairouz, et al. Advances and open problems in federated learning. arXiv:1912.04977, 2019.

---

> > ### Comment · AnonReviewer4 · 2020-11-22
> > **Inclined to increase the rating, but some concerns remain (November 22 revision)**
> >
> >
> >
> > I thank the authors for their latest revision (November 22) and for their response to my comments. Based on the authors' feedback, I am inclined to increase my rating, but I still have some concerns, especially in light of the comments of Reviewer 1 and Reviewer 2.
> >
> > ---
> > ***In relation to Point 4.***
> >
> > > We have read the referenced work. We briefly list the checking results as follows ...
> >
> > **Feedback**
> >
> > I appreciate the detailed response. However, I cannot find any of this discussion in the revision.
> >
> > ---
> >
> > ***In relation to Point 5.***
> >
> > I agree that the authors are just starting out on the asynchronous Byzantine learning problem and it is therefore understandable if the results are not that strong. But it is important to bring out such issues in a paper. Speed-up (wall clock speed-up; statistical learning rate speed-up, etc.) is an important aspect of distributed learning and much more is now known about such speed-ups since the vanilla (A)SGD analysis under the PS model. Two recent references in this direction include:
> >
> > - Advances in Asynchronous Parallel and Distributed Optimization (Assran et al. 2020)
> > - Scaling-up Distributed Processing of Data Streams for Machine Learning (Nokleby et al. 2020)
> >
> > The discussion the authors have added to the revision does not address this issue in a way that informs the reader about this critical point of speed up in distributed learning.
> >
> > ---
> >
> > ***In relation to Point 6.***
> >
> > Advances in Asynchronous Parallel and Distributed Optimization (Assran et al. 2020) is a good reference for this point. It might be useful to have a discussion about this in the paper.
> >
> > ---
> >
> > ***Reviewer 1's Comment (Bounded Gradient Assumption)***
> >
> > The bounded gradient assumption is indeed quite a strong one and is equivalent to being in a local neighborhood of a stationary point. While it is ok for this assumption to make its way into the paper, it is important in my opinion to clarify this limitation of the theoretical analysis in the paper upfront (and not leave it up to the reader to realize this).
> >
> > ---
> >
> > ***Reviewer 2's Comment (Lack of Convergence to a Stationary Point)***
> >
> > I feel the authors' response to this comment of Reviewer 2 is not very satisfactory.
> >
> > - First, the authors can only guarantee that the gradients produced by the algorithm converge to a ball (and not a point), but this is not made obvious in the paper. Some changes in the paper to bring this issue out explicitly would be helpful to general readers of the paper.
> > - Second, the authors claim the size of the ball is very small. Is it possible for the authors to quantify the size of this (gradient) ball for iteration $T$ only to help us see why this should be true? I cannot figure out why the non-decaying additive term should necessarily be a "relative small" positive value.
> >
> > ---

---

> > > ### Author Response · Authors · 2020-11-23
> > > **Reply: We address the remained concerns in the latest version**
> > >
> > > We greatly thank the reviewer for their patience and further comments on our work. All of the three recommended works have been added to the references, since they help us a lot for the improvement. We address the remained concerns in the latest revised version, and hope that the reviewer could re-evaluate our work in light of the improvement.
> > >
> > > ------
> > > ***In relation to Point 4.***
> > > + We have added the discussion after Definition 5. Though, we leave out some details for space-saving. We bring the detailed contents here, as follows:
> > >
> > > "As revealed in (Yang et al., 2020), there are many existing asynchronous Byzantine learning methods. Krum, median, and trimmed-mean are proved to satisfy these two properties (Blanchard et al., 2017; Yin et al., 2018). SignSGD (Bernstein et al., 2019) can be seen as a combination of 1-bit quantization and median aggregation, while median satisfies the properties. Bulyan (Guerraoui et al., 2018) uses an existing aggregation rule to obtain a new one, and the property of Bulyan is difficult to be analyzed alone. Zeno (Xie et al., 2019) has an asynchronous version called Zeno++ (Xie et al., 2020), and it is meaningless to check the properties for Zeno."
> > >
> > > ------
> > > ***In relation to Point 5 & Point 6.***
> > > + We greatly thank the reviewer for the recommended references. Discussion is added at the end of Section 4. We bring the detailed contents here, as follows:
> > >
> > > "As many existing works have indicated (Assran et al., 2020; Nokleby et al., 2020), speed-up is also an important aspect of distributed learning methods. In BASGD, different workers can compute gradients concurrently, make each buffer be filled more quickly, and thus speed up the model updating. However, we mainly focus on Byzantine-resilience in this work. Speed-up will be thoroughly studied in future work. Besides, heavily delayed workers are considered as Byzantine in the current analysis. We will analyze heavily delayed worker's behavior more finely to obtain better results in future work."
> > >
> > > ------
> > > ***Comments of Reviewer R1 and R2.***
> > >
> > > We find that there is a strong connection between the two comments. We will address the two concerns together.
> > > + We thank the reviewer for inspiring us to re-think the problem. The upper-bounded gradient does not guarantee the convergence to a ball. We illustrate this by giving a simple $1$-dimensional counter-example. Let $f(x)=0.01\cos(x)$. Even if we have $|f'(x^t)|\leq 0.01$, we still have no idea about where $x^t$ locates more than $\mathbb{R}$. Intuitively speaking, even if $||\nabla F(w^t)||$ is upper-bounded, $w^t$ may locate in a large area where gradient's norm is small. $w^t$ can also move among different local minima or stationary points. Thus, the upper-bounded gradient only guarantees that a point with a 'relatively small' gradient can be found. To the best of our knowledge, there are hardly much stronger theoretical results for general non-convex cases in existing works. Therefore, we think the relatively weak results are acceptable. We have briefly clarified this limitation of the theoretical analysis after Theorem 2.
> > >
> > > + Besides, we add a proposition after Theorem 2 in the latest revised version. The new proposition guarantees that $A_1\leq D^2$ under a mild condition, meaning that the result of Theorem 2 is stronger than the result directly obtained by the bounded-gradient assumption. We hope that the new proposition could address the concern.

---

> > > > ### Comment · AnonReviewer4 · 2020-11-23
> > > > **A couple of additional clarifications sought ...**
> > > >
> > > > I appreciate the authors' response and the latest revision. I have a couple of additional comments, which I hope can be addressed before the system closes for discussion with the authors.
> > > >
> > > > - Is there a particular reason the authors have not responded to Reviewer 1's comments explicitly?
> > > >
> > > > - So it is correct to assume that indeed the gradient within the ball in which the final solution lies cannot be guaranteed to be too small? I am trying to parse the meaning of _'relatively small' gradient_ and hence this query, since I am unable to see (as noted earlier) why the gradient should be too small.

---

> > > > > ### Author Response · Authors · 2020-11-24
> > > > > **Reply**
> > > > >
> > > > > We greatly thank the reviewer for the interest and appreciation of our work.
> > > > >
> > > > > + We have submitted the response to Reviewer 1. We reply late just because we find the concerns raised by Reviewer 1 is of great importance but a bit tough to deal with. We were stuck on the problem for a long time, but fortunately, get over it at last. We thank Reviewer 4 for additional attention. There is no other reason except for our slow progress. We sincerely apologize if it has caused a misunderstanding.
> > > > >
> > > > > + Briefly speaking, the meaning of 'relatively small' is that $A_1$ is smaller than $D^2$. If loss function $F(w)$ is multiplied by a factor, $A_1$ and $D^2$ will also change. Thus, the exact value of $A_1$ is meaningless. Therefore, we use 'relatively' to mean that $A_1$ is small, compared to some constants that reveal the property of $F(\cdot)$, such as $D^2$. We are sorry for using this ambiguous statement.
> > > > >
> > > > > + Besides, we are not sure about the exact meaning of the sentence "So it is correct to assume that indeed the gradient within the ball in which the final solution lies cannot be guaranteed to be too small?" And we could not give a direct answer to this question. We would greatly appreciate it if the reviewer could explain more about this statement.

---

> > > > > > ### Comment · AnonReviewer4 · 2020-11-24
> > > > > > **Clarifying the last point ...**
> > > > > >
> > > > > > > Besides, we are not sure about the exact meaning of the sentence "So it is correct to assume that indeed the gradient within the ball in which the final solution lies cannot be guaranteed to be too small?" And we could not give a direct answer to this question. We would greatly appreciate it if the reviewer could explain more about this statement.
> > > > > >
> > > > > > Apologies for the not-so-clear-wording. It is my understanding that the authors are guaranteeing convergence to a region in which the gradient does not vanish, but rather is small. But is this "small" just a relative size and in reality one cannot guarantee that it would be **very** small?

---

> > > > > > > ### Author Response · Authors · 2020-11-24
> > > > > > > **Supplementary explanation about the clarified point**
> > > > > > >
> > > > > > > Thanks for the explanation. We think it is hard to give a general conclusion. As revealed in Theorem 2, the gradient's squared norm can decrease to $A_1$ when $T$ increases. However, there are many factors affecting the value of $A_1$, such as loss function $F(\cdot)$, distribution of training instances across workers, number of buffers, and number of Byzantine workers. Thus, Theorem 2 is a very general result. In some "ideal" cases, $A_1$ can be very small, or even $0$, as discussed after Theorem 2. However, in general cases, Theorem 2 may not guarantee a **very** small $A_1$. A **relatively** small value is guaranteed by Theorem 2 and Proposition 2.

---

### Decision · Program_Chairs · 2021-01-07
**Final Decision**

**Decision:**

Reject

**Comment:**

The authors present BASGD and asynchronous version of SGD that attempts to be robust against byzantine failures/attacks.
The papers is overall well written and clearly presents the results. Some novelty is present as there have been limited work in asynchronous algorithms for byzantine ML.

However, there have been several concerns raised by the reviewers, on which I agree, and they have not been fully addressed:
1) the tradeoff between asynchrony and robustness, as BASGD cannot handle the case of a buffer being straggler, which limits some of the novelty in this work
2) issues with the definition of privacy leakage has not been fully addressed
3) some reviewers mentioned the theoretical results being of limited importance, but arguably this is true for other related work in this area. Perhaps a general criticism is valid as to what is the operational value of the proposed guarantees. That is convergence does not exclude a model that has undesirable properties, eg has bad prediction accuracy for a small subset of tasks.
4) Finally, the motivation of the system model of the paper ( eg storing gradients as opposed to instances) paper is of unclear practical relevance, as was raised by multiple reviewers.

Overall the consensus was that the paper does have merits, however, some of the most major concerns were not properly addressed. This paper can potentially be improved for a future venue.